# Grain Size Distribution and Clay Mineral Distinction of Rare Earth Ore through Different Methods

**Lingkang Chen** [1,2,3,*], **Xiongwei Jin** [3], **Haixia Chen** [2], **Zhengwei He** [1,4,*], **Lanrong Qiu** [3] and **Hurong Duan** [5]

1. College of Earth Sciences, Chengdu University of Technology, Chengdu 610059, China
2. College of Sciences, Guangdong University of Petrochemical Technology, Maoming 525000, China; chenhaixia1975@126.com
3. School of Resource and Environmental Engineering, Jiangxi University of Science and Technology, Ganzhou 341000, China; xiongweijjx@126.com (X.J.); lrqiu183@163.com (L.Q.)
4. State Key Laboratory of Geohazard Prevention and Geoenvironment Protection, Chengdu University of Technology, Chengdu 610059, China
5. College of Geomatics, Xi'an University of Science and Technology, Xi'an 710054, China; duanhurong@126.com
* Correspondence: lkchen@jxust.edu.cn (L.C.); hzw@cdut.edu.cn (Z.H.); Tel.: +86-028-84078705 (L.C.)

**Abstract:** Although clay mineral content in ion-absorbed rare earth ores is crucial for migrating and releasing rare earth elements, the formation, distribution, and migration of clay minerals in supergene rare earth ores have not been fully understood. Therefore, this study analyzes the characteristics of clay mineral type and content, soil particle size, pH value, leaching solution concentration, and leaching rate. This analysis was performed using different methods, such as regional rare earth mine soil surveys, in situ leaching profile monitoring, and indoor simulated leaching. The results showed that the grain size and volume curve of rare earth ore have unimodal and bimodal shapes, respectively. X-ray diffraction showed the differences in clay mineral types formed by different weathered bedrocks. The principal clay minerals were kaolinite, illite, chlorite, and vermiculite, with their relative abundance varying with parent rock lithology (granite and low-grade metamorphic rocks). In the Ganxian granite weathering profile, the kaolinite content increased from top to bottom. The decomposition of feldspar minerals to kaolinite was enhanced with an increase in the $SiO_2$ content during weathering. The in situ leaching profile analysis showed that the kaolinite content increased initially and then decreased, whereas the illite/mica content exhibited the opposite trend. Under stable leaching solution concentration and leaching rate, clay mineral formation is favored by lower pH. Low pH, low leaching rate, and highly-concentrated leaching solution (12 wt%) resulted in a slow increase in kaolinite content in the upper part of the profile (30 cm). A lower concentration of the leaching solution (4 wt%) resulted in rapid enrichment of kaolinite after 15 days. Low pH, leaching solution concentration, and leaching rate promoted the formation of distinct kaolinite horizons. We suggest that by disregarding other control factors, rare earth recovery of over 90% can be achieved through leach mining with solutions of 8 wt% and a pH of 5 at a leaching rate of 5 mL/min.

**Keywords:** clay minerals; grain size characteristics; in situ leaching; simulated leaching; ion-absorbed type rare earth ore

## 1. Introduction

Rare earth elements (REEs) are 16 chemical elements grouped by their atomic number, and classified as light (LREEs), middle (MREEs), and heavy (HREEs). The weathered crust elution-deposited REE

ores in southern China have drawn much attention because of their abundance of granitic residuum, their simple extraction processes, and their well-distributed composition [1–3].

The migration and enrichment of REEs are controlled by several factors, such as parent rock lithology, pH value, intensity of weathering, and topography [4–6]. Previous studies have shown that, for chemical index of alteration (CIA) values of 65%–85% in granite, clay minerals increase rapidly with an increasing degree of weathering. There is a positive correlation between the loss on ignition (LOI) of 2%–6% in the weathering crust and REE content [7].

Clay minerals have a controlling effect on the migration and release of REE ore. The completely weathered layer of a weathering crust mainly comprises quartz, feldspar, and clay minerals. The clay mineral content decreases gradually from the weathering crust surface to the lower layer, where clay minerals are converted from hydromica and montmorillonite to halloysite, kaolinite, and gibbsite [8].

The distributions of REEs in the weathering crust are controlled by both the composition of the parent rock and the clay mineral content of the weathering crust. Halloysite, a clay mineral, plays a significant role in the differentiation of cerium [9]. Halloysite has a stronger effect than kaolinite in the adsorption of REEs; however, this adsorption mechanism is not yet fully understood. Previous studies found that the adsorption of REEs is controlled by the properties of the clay minerals rather than the electrolyte solution or dissolved carbon dioxide content [10]. The adsorption capacity of kaolinite increases linearly with increasing pH. A fractionation between HREEs and LREEs due to selective sorption is observed, with HREEs being more sorbed than LREEs at high ionic strengths [10]. For montmorillonite at pHs below 4.5, the REE adsorption capacity is constant, and is modeled by cation exchange [11]. Different clay minerals have different adsorption capacities for REEs. Chi et al. [12] showed that for three common clay minerals, the cation adsorption capacity follows the order: montmorillonite > halloysite > kaolinite. This result shows that different parent rock lithologies will result in different weathering crust structures and clay mineral compositions. Intimate grain-to-grain contacts promote a unique chemical environment at the microscale, bringing about the formation of transient clay mineral phases which quickly disappear in the overlying soil [13]. The bulk of illite in the weathering crust is due to the weathering of mica minerals. A study of unstable soil profiles found that illite is converted into vermiculites or interstratified illite-smectite [14].

Climatic and environmental change is one of the causes of compositional differentiation in clay minerals. Kaolinite and kaolinite interlayer minerals are dominant in strongly leached soil layers [15], while illite and montmorillonite represent a cold and humid climate with weak chemical weathering [16]. Clay minerals of different crystal characteristics differ in physical structure and properties [13]. Clay minerals that host ion-adsorbed REE ores have large specific surface areas and a strong capacity to adsorb REE ions. The clay mineral content thus controls the migration and enrichment of REEs—processes of great significance for REE mineralization.

Although the clay mineralogy of weathered crust elution-deposited REE ores varies, several studies have demonstrated that the clay minerals in these ores commonly comprise halloysite, kaolinite, some illite, and rare montmorillonite [17]. It is widely believed that the horizon enriched in REE generally contains abundant halloysite and kaolinite [18,19], and that clay mineral migration is controlled by soil particle size and specific leaching conditions [20,21]. The metallogenetic mechanism of weathered crust elution REE deposits could involve the weathering of granodiorite and volcanic rocks in warm and humid climates, with the transformation of their parent mineralogy into kaolinite, halloysite, and montmorillonite [22]. In weathering crust elution-deposited REE ores, REEs adsorbed on the clay minerals by ion-exchangeable phases account for more than 80% of the total REE content [3]. However, leaching is controlled by the properties of the REE ore, by the nature and concentration of the leaching reagent, and by the hydrodynamics, kinetics, and mass transfer of the leaching process [22]. We postulate that the weathered crust elution-deposit REE ore is associated with REE ion enrichment, which is dissociated with hydrated or hydroxyl hydrated minerals and adsorbed by clay minerals, which are subsequently deposited, and mineralized in the weathered crust over a long period. In contrast, this is not to say that all REE mineralization can be explained by a single model.

It is important to understand that the clay mineralogy in different environments of REE ore formation varies with different conditions of parent rock, pH values, degrees of weathering [4,5], and mining conditions [9]. This study investigates the types and changing characteristics of clay minerals in several ion-absorbed REE ores in southern Jiangxi Province, China, during weathering and in situ leaching, with an aim of improving the recovery rate. To achieve this, soils on the surface of the weathered crust in a typical rare earth mining area in southern Jiangxi Province were sampled. Then, methods such as in situ leaching profile monitoring and indoor leaching simulation experiments were used to study the characteristics of the clay mineral properties and soil particle size.

## 2. Background

### 2.1. Study Site

The study sites are located in the REE mining regions of Longnan County, Anyuan County, Ganxian District, Ganzhou, Jiangxi Province, China (Figure 1). The region is situated in the subtropical monsoon climatic zone, with an average annual precipitation of 1461.2 mm and an annual average temperature of 19.4 °C [23]. The topography is high in the south and east and low in the north and west [23]. The central region consists mostly of basins between hills. REE mines are mainly distributed in these basins.

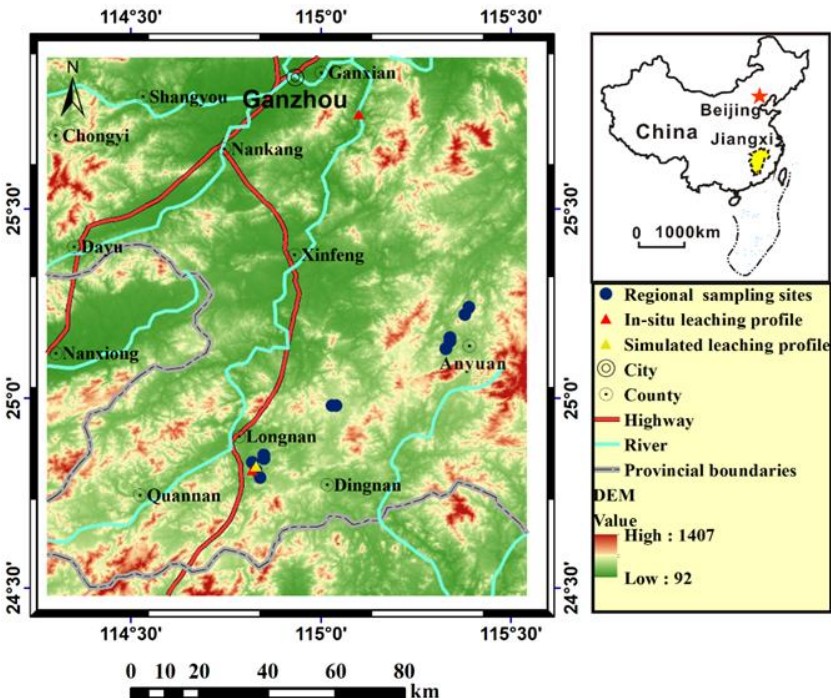

**Figure 1.** The location of sampling sites.

### 2.2. Regional Stratigraphy and Lithology

The study area mainly comprises sandstones, slates, phyllites, and carbonates of the Lower Paleozoic (Sinian) to Mesozoic ages, covered by Quaternary sediments, except for the Silurian, Ordovician, and Tertiary. Quaternary sediments consist of loose fluvial deposits in a river terrace with a high ratio [24].

Magmatic activity in the area can be resolved into four cycles: Caledonian, Variscan, Indosinian, and Yanshanian. The main lithology of igneous intrusions is an acidic to medium-acidic rock [25]. These three types of magmatic rock account for 99.2% of all magmatic rocks in the region. This corresponds to the diagenesis times of 461–384, 228–242, and 189–94 of REE ore-bearing rocks, respectively, as given in [26].

## 3. Materials and Methods

### 3.1. Sample Collection

A total of 49 surface soil samples were collected from REE mines; among these, 27 were sampled regionally, mainly from Anyuan, Longnan, and Dingnan. Ten profile samples were also collected from Jiangwozi, in Ganxian District [27] and from the Wenlong REE mine in Longnan County (longitude 114°49′31.99″E; latitude 24°49′21.68″N).

The in situ leaching profile of the Wenlong REE mine was sampled on three different occasions, October 2, 2016; January 13, 2017; and April 13, 2017, over 193 days (12 samples in total). These samples were collected at elevations from 255 to 249 m and at depths from 0.5 to 1.5 m from the surface; the location was mapped with GPS. The sampled weathering profile comprises, from the surface to the bedrock, purple clayey-sandy soil, red silty clay, gray yellow clayey- sand. The underlying bedrock lithology comprises siliceous slate with high organic content, fine sandstone, grayish-white medium-crystalline granodiorite, fine-medium biotite granite, and grayish-white medium-crystalline granodiorite.

### 3.2. Simulated Leaching Experiment

An unexploited weathering crust with soil horizons similar to those of the in situ leaching profile (i.e., completely weathered upper layer; the transitional middle layer; only partly weathered bottom layer) was selected as representative of the textural and structural characteristics of the typical ion-absorbed REE profile. In the laboratory, a custom column device consisting of a liquid injection barrel, soil column tube, and liquid collection basin was used for the leaching experiment.

The soil column tube used a PVC pipe with a height of 120 cm and inner diameter of 11 cm. An array of sampling holes located at 30, 50, 70, 90, and 110 cm from top to bottom were punched through the pipe wall. The tube bottom was sealed by a lid with a floor drain and plug. To prevent leaching of the sample with the liquid discharge, a paper filter of cotton fiber with 11 cm diameter, similar to the PVC pipe, was laid on the floor of the drain, and the bottom lid was perforated and connected to a plastic hose to receive the leaching ore concentrate. The field profile was sampled following its pedostratigraphy, and the samples were placed in the soil column tube to recreate this stratigraphy, from bottom to top layers. Deionized water spraying was used regularly to ensure that the water content and water holding rate in the soil column sample were approximately consistent with those in the field profile. A total of eight soil columns (T1–T8) were made, each tube was filled with 100 cm of soil, and filter paper with a diameter of 11 cm was placed on the top. The leaching solution was made from analytically pure $(NH_4)_2SO_4$ crystals dissolved in deionized water (from AK-RO-UP-500), to simulate leaching and rainfall. HCl was used to adjust the pH value of the solution. The leaching time was 40 days, with samples taken on the 5th, 15th, 23rd, and 40th day. The machine standard was RO < 0.7 mV and UP < 20 MΩ. The specific leaching parameters of each soil column are shown in Table 1.

**Table 1.** Experimental parameters of simulated leaching.

| Soil Column No. | Leaching Solution | pH | Content of $(NH_4)_2SO_4$ (wt%) | Leaching Solution Flow Rate (mL/min) |
|---|---|---|---|---|
| T1 | $(NH_4)_2SO_4$ | 5 | 8 | 3 |
| T2 | $(NH_4)_2SO_4$ | 3 | 8 | 3 |
| T3 | $(NH_4)_2SO_4$ | 4 | 8 | 3 |
| T4 | $(NH_4)_2SO_4$ | 5 | 12 | 3 |
| T5 | $(NH_4)_2SO_4$ | 5 | 4 | 3 |
| T6 | $(NH_4)_2SO_4$ | 5 | 8 | 1 |
| T7 | $(NH_4)_2SO_4$ | 5 | 8 | 5 |
| T8 (Simulated rainfall) | Deionized water | 6.8~7.2 | 0 | 3 |

### 3.3. Particle size Analysis

The particle size analysis was conducted using different analytical testing methods. Regionally collected samples (27) were processed by the wet sieving method [27]. Particles of less than 0.075 mm were tested on an LS908 (A) laser particle size analyzer (Henan Zhengzhou North-south Instrument Equipment Co. LTD, Zhengzhou, China) at the Jiangxi University of Science and Technology School of Resource and Environmental Engineering. The coarser (>0.075 mm) fraction of the in situ leaching profile was also analyzed by wet sieving, while the <0.075 mm fraction was analyzed with a Malvern MasterSizer 2000 laser particle size analyzer at the Peking University School of Urban and Environmental Sciences.

### 3.4. Clay Mineral Analysis

The XRD analysis of clay minerals followed standard procedures, as described in [28,29]. Bulk samples were pulverized to a fine powder using a planetary ball mill with agate elements. Specimens for XRD analysis were front-loaded using a blade; sieve rotation ensured random grain orientation. The clay fraction was separated in deionized water; the clay suspension was then deposited onto 0.45 μm Whatman filters in vacuum, and transferred to glass slides. Each concentrated clay sample was air-dried before XRD analysis, and then saturated with ethylene glycol for subsequent analysis. Occasionally, heat treatment was necessary; in this case, the slides were heated for one hour at 550 °C before further XRD analysis. Analyses were performed with Rigaku D/max 2550 XRD at the Oil and Gas Laboratory in ALS Houston, USA, and at the Key Laboratory of Nonferrous Metal Materials Science and Engineering of the Ministry of Education, in the Central South University, China. A Bruker Endeavor D4 XRD (Cu radiation, 40 kV, 40 mA) at 0.02564 degree/step/second was used for bulk analysis; clay analysis was performed at 0.02992 degree/step/second.

XRD patterns of separated clay fractions on glass were used for clay mineral identification. Mineral identification was facilitated by JADE (version 9.5). Quantitative analysis of minerals was performed by the Rietveld method [30], and amorphous phases were not accounted. The results were normalized to 100% based on the assumption that the complete mineral content of the sample was accounted for in the XRD patterns. Duplicate samples were analyzed at the two laboratories in the United States and China, and the error was less than 10%.

## 4. Results

### 4.1. Particle Size Analysis

A total of 27 regional particle size samples were analyzed. Particle size distribution curves from this sample set show both unimodal and bimodal patterns (Figure 2).

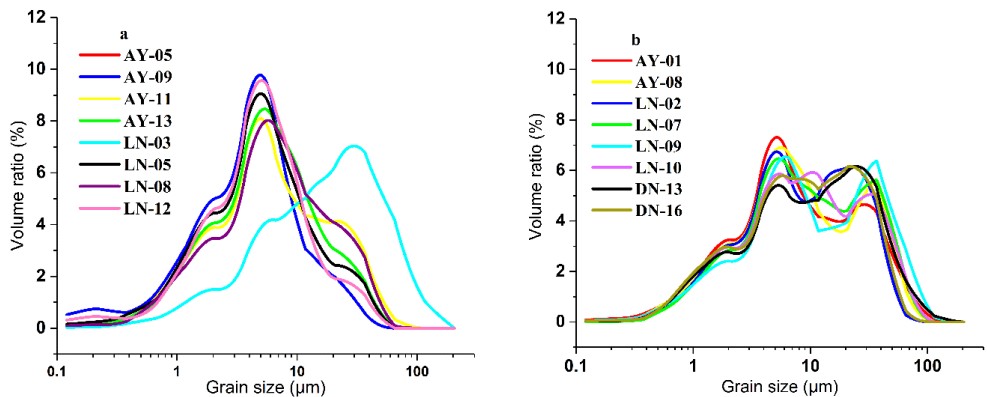

**Figure 2.** Grain size distribution of 27 regional samples: (**a**) shows unimodal patterns; and (**b**) shows bimodal patterns.

　　　Particle size curves with unimodal patterns presented approximate normal distribution, with the exception of Sample LN-03 (Figure 2a). A significant peak was estimated in the particle size range of 2.1–17.2 µm, with volume ratio from 8.96 to 10.36. The range of grain size ratio was more than 70% and was controlled by fine particles. Significantly, Sample LN-03 showed a unimodal curve with obvious right-hand deviation (Figure 2a), with a grain size peak at 30.5 µm and volume ratio of 7.2%. Therefore, in Sample LN-03, the peak volume ratio was lower and the particle size was coarser than that in other unimodal samples. In samples of bimodal particle size distribution, the first significant peak appeared at 1.7 µm (Figure 2b), with an estimated particle size range of 1.4–2.6 µm and volume ratio of 1.5%–5%. The maximum peak interval was in the particle size range of 2.6–9.7 µm, whereas the volume ratio was in the range of 1.47%–10.07%. The third peak interval was in the particle size range of 14.2–36.9 µm, with a volume ratio of 4% to 7.5%.

　　　Samples with unimodal particle size distribution (Figure 2a; e.g., AY-05 and AY-09) originated in weathering crust comprising red silty clay, indicative of advanced bedrock weathering and pedogenesis. This weathering crust overlies bedrock lithology with siliceous slate and fine sandstone interlayers. In contrast, samples with bimodal particle size distribution originated in soils over grayish-white medium-grained granodiorite with moderate weathering.

　　　In the samples from the weathering crust profile, the grain size increased gradually up-profile, reflecting the degree of weathering in different profile horizons. Sample LN-03, with a coarser particle size overall (Figure 2a), came from a depth of 0.90 m, which is close to party weathering bedrock. The particle volume distribution curves indicate that particle size distribution in the REE mine does not follow a single normal distribution pattern, thus reflecting considerable differences in soil particle size gradation. These analyses show that sample location, degree of weathering, and bedrock lithology affect the distribution of soil particle size.

## 4.2. Regional Clay Mineral Analysis

　　　The clay minerals in the regional samples were mainly kaolinite, followed by illite, chlorite, vermiculite (Table 2). Mineral residues from the parent rock included quartz and potassium feldspar with small quantities of plagioclase in some samples. The kaolinite content varied greatly in the regional samples (maximum: 62.1%; minimum: 8.8%; average: 31.91%; standard deviation of 11.90).

**Table 2.** Main clay minerals content of regional in soil.

| Serial Number | Sample | Clay (%) | | | | Other Minerals (%) | | |
|---|---|---|---|---|---|---|---|---|
| | | Vermiculite | Chlorite | Kaolinite | Illite-Mica | Quartz | Potassium Feldspar | Plagioclase |
| 1 | AY-01 * | - | 1.8 | 37 | 9.4 | 42.8 | 8.3 | - |
| 2 | AY-02 | - | - | 40.3 | 10.1 | 39.8 | 9.8 | - |
| 3 | AY-03 * | - | - | 37.5 | 13.5 | 40.8 | 6.7 | - |
| 4 | AY-05 * | - | 2.8 | 37.3 | 3.4 | 38 | 13.4 | - |
| 5 | AY-06 * | - | 2.7 | 38.1 | 6.3 | 30.8 | 15.9 | - |
| 6 | AY-07 | - | - | 62.1 | 13.9 | 20.3 | 3.7 | - |
| 7 | AY-08 | - | - | 56 | 16.4 | 22.8 | 4.7 | - |
| 8 | AY-09 * | - | 2.5 | 31.5 | 5.9 | 42.2 | 10.5 | - |
| 9 | AY-10 | - | - | 36.9 | 9.5 | 42.2 | 11.3 | - |
| 10 | AY-11 | - | - | 36.2 | 4.9 | 55.7 | 3.2 | - |
| 11 | AY-12 * | - | 3.4 | 29.2 | 0 | 43.8 | 6 | - |
| 12 | AY-13 * | 6.6 | 1 | 45.8 | 3.1 | 23.3 | 23.1 | - |
| 13 | DN-13 * | - | 2.6 | 28.6 | 1.8 | 31.4 | 31.8 | - |
| 14 | DN-16 * | 1.5 | 1.5 | 24.5 | 1.3 | 34.1 | 35.3 | - |
| 15 | LN-01 * | 1.9 | 0.9 | 32.8 | - | 48.7 | 6.8 | - |
| 16 | LN-02 | 3.8 | - | 25.4 | 21 | 35 | 18.7 | - |
| 17 | LN-03 | - | - | 8.8 | 28.6 | 54.4 | 8.2 | - |
| 18 | LN-04 * | - | - | 30.3 | 2 | 47.8 | 18.6 | - |
| 19 | LN-05 | 0.1 | - | 23 | 19.1 | 44.1 | 13.8 | - |
| 20 | LN-06 * | | - | 11.9 | 0.4 | 31.5 | 29.3 | 10 |

**Table 2.** *Cont*.

| Serial Number | Sample | Clay (%) | | | | Other Minerals (%) | | |
|:---:|:---:|:---:|:---:|:---:|:---:|:---:|:---:|:---:|
| | | Vermiculite | Chlorite | Kaolinite | Illite-Mica | Quartz | Potassium Feldspar | Plagioclase |
| 21 | LN-07 | 1.5 | - | 12.4 | 13 | 59.6 | 15 | - |
| 22 | LN-08 * | - | - | 30 | 0.3 | 47.8 | 13.2 | 0.4 |
| 23 | LN-09 * | 1.4 | - | 33.7 | 0.6 | 43.7 | 16.9 | - |
| 24 | LN-10 * | 0.6 | - | 37.9 | 1.3 | 39.3 | 15.4 | 1.2 |
| 25 | LN-11 | 0.8 | - | 13.5 | 6.4 | 53.9 | 26.2 | - |
| 26 | LN-12 * | - | - | 30.1 | 0.3 | 38.3 | 25 | 1 |
| 27 | LN-18 * | - | - | 30.9 | 3.2 | 49.8 | 6.5 | 1.5 |

\* Sample analyses were supported by Oil and Gas Laboratory in ALS Houston.

Since the chemical composition of kaolinite is the same as that of halloysite (except for weakly-bound interlayer water) [31], the kaolinite diffraction peak is the same as that of halloysite. Fang et al. [9] studied clay minerals in six REE mining areas in southern Jiangxi Province; the characteristic diffraction peaks of halloysite and kaolinite were 7.30–7.45 Å, 4.5–4.6 Å, 3.58–3.60 Å, and 3.32–3.37 Å. Chi et al. [3] found that the weathered crust leaching type REE ore was mainly composed of clay minerals, i.e., mainly halloysite, illite, kaolinite, and very small amounts of montmorillonite, alongside quartz sand and rock-forming minerals (feldspar). Halloysite is generally formed in the upper layer of a weathering crust from kaolinite interstratified minerals in noncrystalline stage, resulting from weathering and the dissolution of feldspar [13,15,32]. We infer that the kaolinite identified through XRD in our samples may contain kaolinite and kaolinite-interstratified minerals. The coexistence of both minerals in weathering profiles is frequently reported in studies with electron microscopy [32,33].

### 4.3. Analysis of Clay Minerals in the in Situ Leaching Profile

Samples from the Jiangwozi and Longnan profiles, Ganxian District, mainly comprised kaolinite and illite (i.e., mica), along with quartz and feldspar (Table 3). The kaolinite content ranged from 8.3% to 35.0% (average: 18.88%; standard deviation: 7.82).

**Table 3.** Main clay minerals content of soil profile (%).

| Serial Number | Sample | Kaolinite | Illite-Mica | Quartz | Potassium Feldspar | Depth (cm) |
|:---:|:---:|:---:|:---:|:---:|:---:|:---:|
| 1 | GX-01 | 16.2 | 19.2 | 39.7 | 24.8 | 20 |
| 2 | GX-02 | 18.7 | 13.5 | 37 | 30.8 | 45 |
| 3 | GX-03-2 | 25.9 | 15.3 | 33.6 | 25.2 | 60 |
| 4 | GX-03-1 | 20.3 | 7.3 | 39.7 | 32.7 | 80 |
| 5 | GX-04-5 | 28.2 | 13.6 | 31 | 27.2 | 105 |
| 6 | GX-04-4 | 14.9 | 15.9 | 34 | 35.2 | 130 |
| 7 | GX-04-3 | 16.4 | 11.3 | 33.3 | 39 | 150 |
| 8 | GX-04-2 | 16.5 | 22.5 | 32.3 | 28.6 | 170 |
| 9 | GX-04-1 | 35 | 17.2 | 33.1 | 14.7 | 190 |
| 10 | GX-05 | 9.8 | 12.4 | 39.9 | 37.9 | 215 |
| 11 | P01-1-1 | 12.6 | 17.2 | 46.2 | 31.3 | 80 |
| 12 | P01-2-1 | 12.3 | 16 | 40.5 | 34.1 | 150 |
| 13 | P01-3-1 | 9.5 | 12.3 | 44 | 24.9 | 195 |
| 14 | P01-4-1 | 11.8 | 10.1 | 53.2 | 16.4 | 280 |
| 15 | P01-1-2 | 8.9 | 13.4 | 61.3 | 25.1 | 80 |
| 16 | P01-2-2 | 8.3 | 13 | 53.6 | 18.3 | 150 |
| 17 | P01-3-2 | 33 | 11.9 | 36.8 | 24.2 | 195 |
| 18 | P01-4-2 | 17.7 | 15 | 43 | 20.9 | 280 |
| 19 | P01-1-3 | 22.5 | 20.7 | 35.8 | 30.6 | 80 |
| 20 | P01-2-3 | 22.8 | 19.9 | 26.7 | 21.4 | 150 |
| 21 | P01-3-3 | 31.2 | 13.8 | 33.6 | 34.3 | 195 |
| 22 | P01-4-3 | 22.8 | 15 | 27.9 | 31.3 | 280 |

In the Jiangwozi profile, the kaolinite content increases nonlinearly from the surface to the bottom, from 16.2% to 28.2% (Table 2), and then decreases gradually. The kaolinite content reached a maximum of 35.0% at 200 cm from the surface, where the soil texture is coarse and fine sand. At this depth, kaolinite is distributed like a network [27]. The REE ore of the Jiangwozi profile is mainly enriched in the weathering and leaching layer between 90 and 200 cm. The change of clay mineral content is closely related to the chemical weathering rate of the rock: the maximum kaolinite content (35.0%) is in GX-04-1 where potassium feldspar content is at its minimum (14.7%), suggesting that potassium feldspar is strongly weathered. At depths of 0–55 cm, the illite (mica) content showed the opposite trend to that of kaolinite. From 70 to 120 cm, illite (i.e., mica) follows the same trend as kaolinite, but its content is lower. At this depth, the content of coarse particles is relatively increased, probably due to enhancement of chemical weathering and vertical migration of particles. The kaolinite content peaks at about 200 cm, while the illite content peaks earlier, at around 190 cm (at 17.2%). Yang [34] researched the clay mineralogy of the REE weathering crust of Longnan granite, in Jiangxi Province, and found that the crystallization degree of kaolinite gradually increased down-profile, suggesting that the content of kaolinite also increased. This discovery was demonstrated in granite weathering crust profiles where kaolinite was dominant at the bottom of crust [13,35].

The analysis of soil clay minerals and rare earths in the Wenlong mine showed that at the onset of in situ leaching, kaolinite is absent; however, as leaching continues, the kaolinite content increases dramatically at depth of around 200 cm (Table 2). In comparison, with the significant change with depth in the Ganxian District profile, at the Wenlong mine profile, kaolinites formed in the course of weathering increase gradually as in situ leaching progresses. This is the result of the coupling of natural weathering and human activity (profile stripping in the course of REE mining).

### 4.4. Clay Minerals in the Simulated Leaching Profile

The clay (kaolinite and illite) and other mineral (quartz, potassium feldspar, and plagioclase) content of the eight soil profiles subjected to the simulated leaching experiment are presented in Table 4. Each soil column sample is denoted by a three-digit number: the first digit (T1 to T8) represents the number of the soil column; the second digit (1, 3, and 5) represents the depth of the sample in the soil profile (at 30, 70, and 110 cm, respectively); and the third digit (1, 2, 3, and 5) represents the time of mineral concentration measurement since the onset of the experiment (5th, 15th, 23rd, and 40th day, respectively).

**Table 4.** Clay and other mineral content of eight soil profiles (T1 to T8) at different times during simulated leaching (%).

| Serial Number | Sample | Kaolinite | Illite-Mica | Quartz | Potassium Feldspar | Plagioclase |
|:---:|:---:|:---:|:---:|:---:|:---:|:---:|
| 1 | T1-1-1 | 22.3 | 7.4 | 45.7 | 24.6 | - |
| 2 | T1-1-2 | 21.5 | 7.8 | 48.1 | 22.6 | - |
| 3 | T1-1-3 | 20.2 | 5.1 | 45.2 | 29.6 | - |
| 4 | T1-1-5 | 11.1 | 12.9 | 45.9 | 30.1 | - |
| 5 | T1-3-1 | 18.9 | 8.8 | 55.6 | 16.7 | - |
| 6 | T1-3-2 | 14.9 | 9.4 | 39.4 | 36.2 | - |
| 7 | T1-3-3 | 13.8 | 13.2 | 51.7 | 21.3 | - |
| 8 | T1-3-5 | 6.8 | 14.7 | 54.6 | 23.9 | - |
| 9 | T1-5-1 | 15.2 | 14.6 | 36.6 | 33.6 | - |
| 10 | T1-5-2 | 13.1 | 8.6 | 48.1 | 30.2 | - |
| 11 | T1-5-3 | 16.1 | 12.8 | 37.2 | 33.9 | - |
| 12 | T1-5-5 | 17.5 | 8.2 | 39.2 | 35.1 | - |
| 13 | T2-1-1 | 20.5 | 11.4 | 42.5 | 25.5 | - |
| 14 | T2-1-2 | 16.6 | 12.5 | 37.7 | 33.2 | - |
| 15 | T2-1-3 | 21.7 | 7.2 | 45.3 | 25.8 | - |
| 16 | T2-1-5 | 20.2 | 10.2 | 46.3 | 23.2 | - |

**Table 4.** *Cont.*

| Serial Number | Sample | Kaolinite | Illite-Mica | Quartz | Potassium Feldspar | Plagioclase |
|---|---|---|---|---|---|---|
| 17 | T2-3-1 | 11.2 | 12.5 | 44.8 | 35.1 | - |
| 18 | T2-3-2 | 9.2 | 15.0 | 43.0 | 32.8 | - |
| 19 | T2-3-3 | 8.8 | 23.2 | 35.9 | 32.1 | - |
| 20 | T2-3-5 | 10.5 | 23.0 | 43.1 | 23.5 | - |
| 21 | T2-5-1 | 8.9 | 20.8 | 44.4 | 25.8 | - |
| 22 | T2-5-2 | 8.5 | 16.0 | 42.2 | 33.4 | - |
| 23 | T2-5-3 | 25.3 | 1.2 | 45.5 | 28.0 | - |
| 24 | T2-5-5 | 31.5 | 5.3 | 31.0 | 32.2 | - |
| 25 | T3-1-1 | 15.3 | 10.5 | 55.5 | 18.7 | - |
| 26 | T3-1-2 | 16.2 | 12.8 | 40.2 | 30.8 | - |
| 27 | T3-1-3 | 15.7 | 16.9 | 39.6 | 27.8 | - |
| 28 | T3-1-5 | 18.4 | 9.7 | 48.4 | 23.5 | - |
| 29 | T3-3-1 | 9.2 | 17.9 | 43.3 | 29.6 | - |
| 30 | T3-3-2 | 11.2 | 14.2 | 45.7 | 28.9 | - |
| 31 | T3-3-3 | 10.5 | 15.8 | 43.5 | 30.3 | - |
| 32 | T3-3-5 | 18.6 | 20.2 | 38.0 | 23.2 | - |
| 33 | T3-5-1 | 16.5 | 15.0 | 28.5 | 40.1 | - |
| 34 | T3-5-2 | 9.2 | 11.4 | 44.8 | 34.6 | - |
| 35 | T3-5-3 | 11.8 | 21.9 | 28.3 | 38.1 | - |
| 36 | T3-5-5 | 16.2 | 17.5 | 25.9 | 40.5 | - |
| 37 | T4-1-1 | 14.1 | 14.9 | 43.9 | 27.2 | - |
| 38 | T4-1-2 | 18.1 | 12.3 | 48.5 | 21.1 | - |
| 39 | T4-1-3 | 18.2 | 11.8 | 42.8 | 27.2 | - |
| 40 | T4-1-5 | 18.5 | 10.6 | 34.0 | 36.9 | - |
| 41 | T4-3-1 | 9.9 | 12.2 | 46.2 | 31.7 | - |
| 42 | T4-3-2 | 6.9 | 16.1 | 41.8 | 35.2 | - |
| 43 | T4-3-3 | 10.4 | 16.7 | 29.8 | 43.1 | - |
| 44 | T4-3-5 | 15.7 | 20.4 | 37.3 | 26.6 | - |
| 45 | T4-5-1 | 14.3 | 16.5 | 34.0 | 35.2 | - |
| 46 | T4-5-2 | 9.1 | 14.0 | 42.5 | 34.4 | - |
| 47 | T4-5-3 | 15.3 | 9.5 | 50.1 | 25.1 | - |
| 48 | T4-5-5 | 11.2 | 6.8 | 51.4 | 27.4 | 3.1 |
| 49 | T5-1-1 | 14.2 | 9.6 | 48.3 | 27.9 | - |
| 50 | T5-1-2 | 16.9 | 14.0 | 43.6 | 25.5 | - |
| 51 | T5-1-3 | 16.4 | 30.6 | 35.4 | 17.6 | - |
| 52 | T5-1-5 | 26.7 | 9.6 | 34.2 | 29.5 | - |
| 53 | T5-3-1 | 16.6 | 8.9 | 49.4 | 25.2 | - |
| 54 | T5-3-2 | 12.4 | 12.1 | 47.9 | 27.5 | - |
| 55 | T5-3-3 | 16.2 | 11.1 | 41.5 | 31.2 | - |
| 56 | T5-3-5 | 12.1 | 17.1 | 39.8 | 31.0 | - |
| 57 | T5-5-1 | 15.2 | 10.2 | 29.0 | 37.8 | 7.9 |
| 58 | T5-5-2 | 9.4 | 8.8 | 31.8 | 32.4 | 17.6 |
| 59 | T5-5-3 | 11.4 | 13.2 | 36.7 | 29.7 | 9.1 |
| 60 | T5-5-5 | 9.6 | 11.5 | 54.2 | 24.6 | - |
| 61 | T6-1-1 | 10.9 | 6.9 | 52.9 | 23.9 | 5.4 |
| 62 | T6-1-2 | 10.5 | 8.4 | 57.7 | 19 | 4.4 |
| 63 | T6-1-3 | 9.8 | 8.6 | 57.7 | 23.9 | - |
| 64 | T6-1-5 | 14.0 | 12.5 | 46.9 | 26.6 | - |
| 65 | T6-3-1 | 13.3 | 13.2 | 40.6 | 30.4 | 2.4 |
| 66 | T6-3-2 | 15.1 | 12.9 | 49.7 | 22.3 | - |
| 67 | T6-3-3 | 13.7 | 10.3 | 47.0 | 29.0 | - |
| 68 | T6-3-5 | 13.3 | 12.8 | 60.6 | 13.2 | - |
| 69 | T6-5-1 | 9.8 | 13.9 | 47.8 | 20.8 | 7.7 |
| 70 | T6-5-2 | 7.0 | 11.6 | 59.1 | 22.3 | - |
| 71 | T6-5-3 | 15.4 | 12.9 | 40.8 | 26.1 | 4.8 |
| 72 | T6-5-5 | 16.4 | 12.3 | 38.0 | 29.8 | - |
| 73 | T7-1-1 | 16.8 | 10.2 | 43.0 | 22.5 | 7.4 |
| 74 | T7-1-2 | 15.7 | 13.2 | 40.1 | 31.0 | - |
| 75 | T7-1-3 | 20.6 | 13.3 | 39.9 | 26.2 | - |
| 76 | T7-1-5 | 19.1 | 21.3 | 38.3 | 21.3 | - |

**Table 4.** *Cont.*

| Serial Number | Sample | Kaolinite | Illite-Mica | Quartz | Potassium Feldspar | Plagioclase |
|---|---|---|---|---|---|---|
| 77 | T7-3-1 | 7.5 | 15.5 | 40.5 | 30.5 | 6.1 |
| 78 | T7-3-2 | 10.3 | 13.1 | 51.3 | 22.4 | 2.9 |
| 79 | T7-3-3 | 15.3 | 4.5 | 38.5 | 31.1 | 10.6 |
| 80 | T7-3-5 | 11.3 | 10.9 | 53.3 | 19.6 | 4.9 |
| 81 | T7-5-1 | 13.5 | 8.0 | 34.1 | 38 | 6.4 |
| 82 | T7-5-2 | 11.0 | 14.6 | 32.8 | 41.6 | - |
| 83 | T7-5-3 | 18.3 | 12.4 | 28.0 | 41.3 | - |
| 84 | T7-5-5 | 25.3 | 17.7 | 41.5 | 15.5 | - |
| 85 | T8-1-1 | 16.0 | 9.6 | 46.4 | 28.0 | - |
| 86 | T8-1-5 | 20.6 | 19.3 | 30.7 | 29.4 | - |
| 87 | T8-3-1 | 8.5 | 5.4 | 34.7 | 51.4 | - |
| 88 | T8-3-5 | 15.2 | 11.5 | 35.7 | 37.6 | - |
| 89 | T8-5-1 | 14.6 | 11.3 | 49.9 | 24.2 | - |
| 90 | T8-5-5 | 19.6 | 17.7 | 33.8 | 28.9 | - |

-: represented that its content was below the detection limit.

The lowest kaolinite content is 6.8%, at T1-3-5, i.e., at sampling port number 3 in the T1 soil column after 40 days of simulated leaching. The highest kaolinite content is 31.5%, at T2-5-5, i.e., sampling port number 5 in the T2 soil column after 40 days of simulated leaching. The average content of kaolinite in the REE ore is 14.67% and the standard deviation is 4.72. In all columns except T1 and T2, the first sampling port (depth: 30 cm) shows that the variation in kaolinite content increases with the longer leaching. This is most obvious in T5, where the kaolinite content after 40 days of simulated leaching (sample No: T5-1-5) is 1.88 times the initial value. In soil columns T1 and T2, the kaolinite content changed little in the first 23 days, but after 40 days, it declined significantly in T1. The highest illite content is 30.6%, at T5-1-3, i.e., sampling port number 1 in the T5 soil column after 23 days of simulated leaching. Overall, the average illite content in the soil column sample set is 12.78%, with a standard deviation of 4.63, slightly less than that of kaolinite. Similarly, the illite-mica content of the first sampling port (depth of 5 cm) increased gradually in T1 and T6–T8 as leaching progressed, while other soil columns showed no obvious variation. However, in the T5 soil column, the illite-mica peaked at 30.6% after 23 days of leaching, and then fell back to the initial level after 40 days of leaching. The second and third sampling ports show that the content of kaolinite, illite and other clay minerals in the RE ore is complex under different leaching conditions (Table 3), which may be controlled by many factors.

The minimum content of potassium feldspar is 13.2%, after 40 days of leaching (sample No: T6-3-5), which is 56.6% lower than the potassium feldspar content at the sampling port at the initial stage of leaching (sample No: T6-3-1: 30.4%). This result indicates that potassium feldspar at the bottom of the soil column may have been weathered and mobilized with the leaching solution after prolonged leaching. The maximum potassium feldspar content is 51.4% (T8-3-1); average value is 28.81%, and the standard deviation is 6.60. Quartz fluctuates from 25.9 to 60.6% in the course of ore leaching, with an average value of 42.6%, and a standard deviation of 7.67, which is greater than that of potassium feldspar. In addition, plagioclase was detected in 15 samples, and its content fluctuated between 2.4% and 17.6%, with an average value of 6.71% and a standard deviation of 3.68. In the other samples, the plagioclase content was below the detection limit.

## 5. Discussion

### 5.1. Soil Particle Size and Distribution of Clay Minerals in REE Mining Areas

Ion-absorbed REE ore is mainly formed by advanced weathering of granite. It is a loose, earthy substance comprising quartz, feldspar, and clay minerals [36]. Therefore, this loose soil mantle, formed

by surface weathering is closely related to mineral grain size. The cumulative curves of regional particle size distribution are S-shaped (Figure 3), which is consistent with earlier report [36].

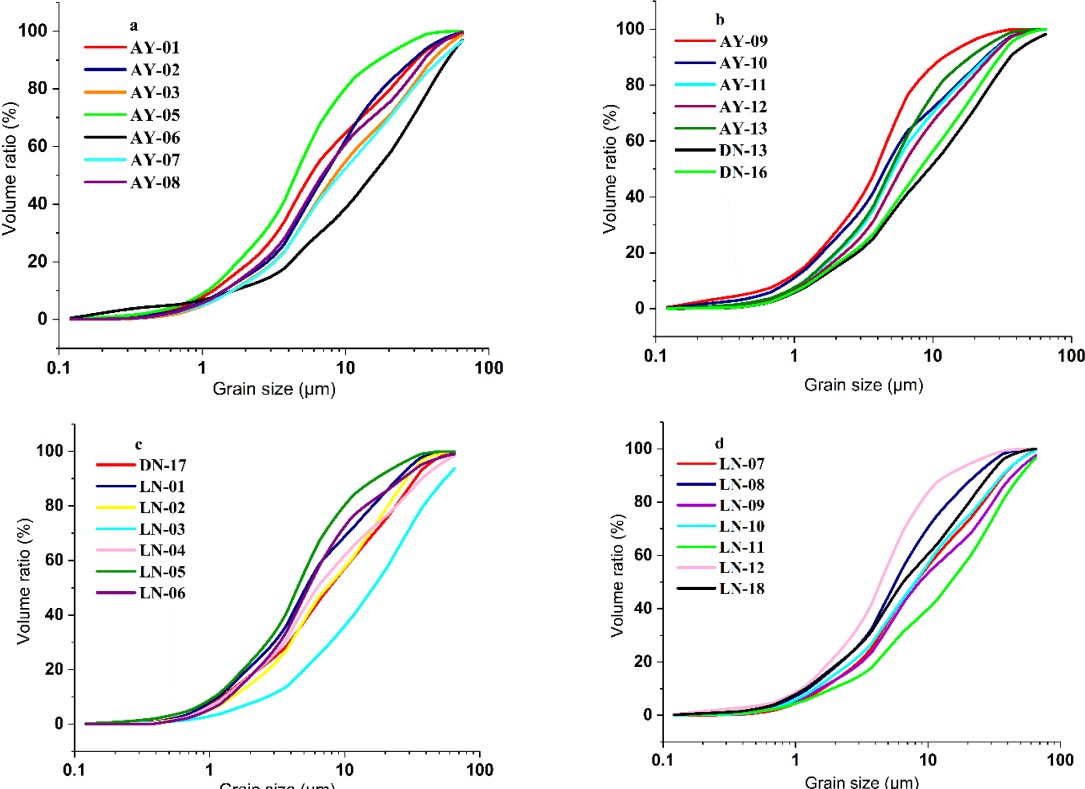

**Figure 3.** Cumulative particle size distribution curves from ion-absorbed RE in (regional samples). (**a**) shows the curves with AY-01, AY-02, AY-03, AY-05, AY-06, AY-07 and AY-08; (**b**) shows the curves with AY-09, AY-10, AY-11, AY-12, AY-13, DN-13 and DN-16; (**c**) shows the curves with DN-17, LN-01, LN-02,LN-03, LN-04, LN-05 and LN-06; (**d**) shows the curves with LN-07, LN-08, LN-09, LN-10, LN-11, LN-12 and LN-18.

Additionally, the particle distribution curves in Figure 3a–d show that particle size ranges between 3.7 and 30.5 μm. According to Aberg's classification of granular materials [36,37], some of the cumulative curves of the regional sample set are A-shaped (i.e., the left end of the cumulative particle distribution curve is relatively steep, with a concave side downward), indicating that the particle gradation changes significantly with an increase in coarse particles in the soil. Yan et al. [38] used a wet sieving method to classify REE ores into eight types of particle size distribution. In this paper, particle size analysis concentrated on the <0.075 mm size fraction, and the analytical method was different (Malvern-2000 laser particle size analyzer); therefore, our particle size distribution curves are different.

The cumulative particle size distribution curves of AY-05 (Figure 3a), AY-09 (Figure 3b), LN-05 (Figure 3c), and LN-12(Figure 3d), nevertheless, are inclined steeply to left in their upper part (a B-type structure in Arberg's terms [37]), indicating that grain size changes abruptly from fine to coarse. A possible reason for this is that simulated leaching continues to promote rapid decomposition of feldspar (Table 3). Dissolution of feldspar due to leaching releases $SiO_2$, which migrates downward, where it recrystallizes resulting in particle thickening [36].

Eigenvalue analysis shows that the 10 particle size (the particle size at which the cumulative particle size distribution curve reaches 10% of the volume) was in the range of 0.82–5.03 μm (Table 5). The minimum D10 value (0.82), corresponds to Sample AY-09, a red sand sampled from 1.5 m below- surface, above Sinian feldspar quartz and slate bedrock in the age of Sinian period (Z). Field investigation in this area revealed that the weathering crust is approximately 3.5–3.8-m thick, and

that the bedrock is strongly deformed and fractured, and thus particularly susceptible to physical and chemical weathering. D90 (the particle size at which the cumulative particle size distribution curve reaches 90% of the volume) ranges from is 11.99 μm (minimum) to 60.99 μm (maximum), with a standard deviation of 11.4, respectively, thus suggesting that discreteness increases with particle size. The average particle size, Dav and volumetric average particle size, D [4,3] have similar maximum, minimum, and standard deviation values. The standard deviation of the median particle diameter, D50 (the particle size at which the cumulative particle size distribution curve reaches 50% of the volume) is more significant than that of D [3,2] (surface area average particle size), indicating that the median particle diameter is more discrete than the surface area average particle size.

**Table 5.** Characteristic parameters of particle size distribution; regional weathering crust samples.

| Sample No. | D10 (μm) | D50 (μm) | D90 (μm) | Dav (μm) | D [3,2] (μm) | D [4,3] (μm) |
|---|---|---|---|---|---|---|
| AY-01 | 1.32 | 6.25 | 33.88 | 12.88 | 3.09 | 12.88 |
| AY-02 | 1.43 | 7.28 | 29.54 | 11.65 | 3.36 | 11.65 |
| AY-03 | 1.64 | 8.47 | 40.28 | 15.64 | 4.09 | 15.64 |
| AY-05 | 1.10 | 4.50 | 17.10 | 7.02 | 2.30 | 7.02 |
| AY-06 | 1.79 | 15.51 | 50.28 | 21.57 | 3.67 | 21.57 |
| AY-07 | 2.21 | 12.76 | 40.04 | 17.22 | 4.75 | 17.22 |
| AY-08 | 1.44 | 6.93 | 35.50 | 13.21 | 3.48 | 13.21 |
| AY-09 | 0.82 | 3.88 | 11.99 | 5.40 | 1.77 | 5.40 |
| AY-10 | 0.96 | 4.57 | 23.70 | 8.76 | 2.21 | 8.76 |
| AY-11 | 1.34 | 5.38 | 20.86 | 8.42 | 2.89 | 8.42 |
| AY-12 | 1.13 | 6.39 | 28.22 | 11.06 | 2.61 | 11.06 |
| AY-13 | 1.20 | 5.02 | 18.49 | 7.70 | 2.58 | 7.70 |
| DN-13 | 1.64 | 12.75 | 44.85 | 18.90 | 4.37 | 18.90 |
| DN-16 | 1.37 | 8.07 | 30.07 | 12.42 | 3.50 | 12.42 |
| LN-01 | 1.16 | 5.14 | 24.69 | 9.41 | 2.84 | 6.10 |
| LN-02 | 1.50 | 7.45 | 28.71 | 11.85 | 3.59 | 11.85 |
| LN-03 | 5.03 | 20.86 | 61.00 | 27.80 | 8.19 | 27.80 |
| LN-04 | 1.49 | 8.39 | 38.06 | 14.92 | 3.67 | 14.92 |
| LN-05 | 1.37 | 6.42 | 33.75 | 13.00 | 3.53 | 13.00 |
| LN-06 | 1.41 | 5.47 | 26.90 | 10.20 | 3.31 | 10.20 |
| LN-07 | 1.62 | 8.17 | 37.10 | 14.42 | 4.07 | 14.42 |
| LN-08 | 1.26 | 5.68 | 22.95 | 9.08 | 2.89 | 9.08 |
| LN-09 | 1.55 | 8.63 | 43.42 | 17.11 | 3.74 | 17.11 |
| LN-10 | 1.41 | 8.08 | 36.38 | 13.98 | 3.58 | 13.98 |
| LN-11 | 1.91 | 15.00 | 49.19 | 21.02 | 4.51 | 21.02 |
| LN-12 | 1.12 | 4.41 | 13.85 | 6.45 | 2.08 | 6.45 |
| LN-18 | 1.21 | 6.68 | 28.74 | 11.42 | 2.79 | 11.42 |

With Dav as an independent variable and the other characteristic parameters as dependent variables (Figure 4), the regression coefficient is D90 > D [4,3] > D50 > D [3,2] > D10. This finding illustrates that the increase in surface average particle size has a more significant impact on coarse particles than on fine particles (D10). D [4,3] has the highest correlation with Dav (correlation coefficient: 0.99102), followed by D90, D50, D [3,2], and the lowest correlation with D10 (correlation coefficient: 0.727).

D10 residual analysis (Figure 5a) shows that when the average particle size Dav increases, other residuals decrease (with very few exceptions). This shows that for D10, as the average particle size increases, the volume of particle grain size decreases by less than 10%. Residuals are normally distributed (Figure 5b), and the regression analysis of the dependent variable is similar to that of the independent variable. Residuals present a linear shape at a 99.5% confidence interval (Figure 5c,d).

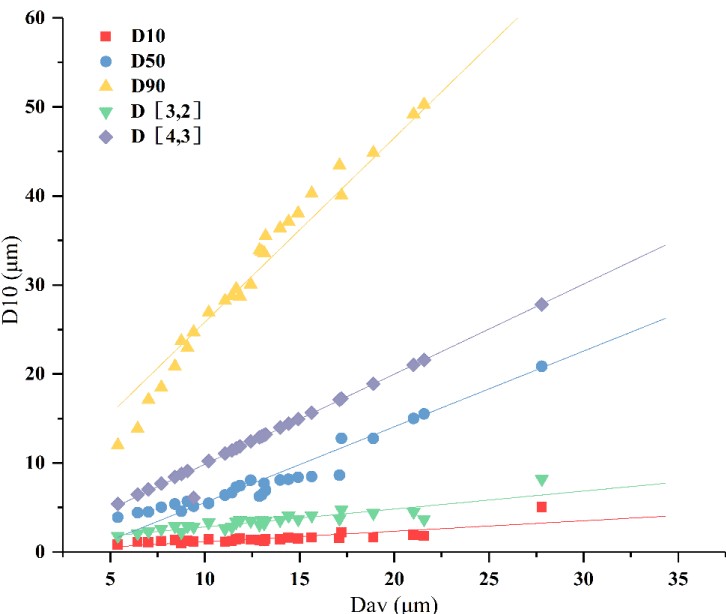

**Figure 4.** Scatterplot of characteristic parameters of particle size distribution; regional weathering crust samples.

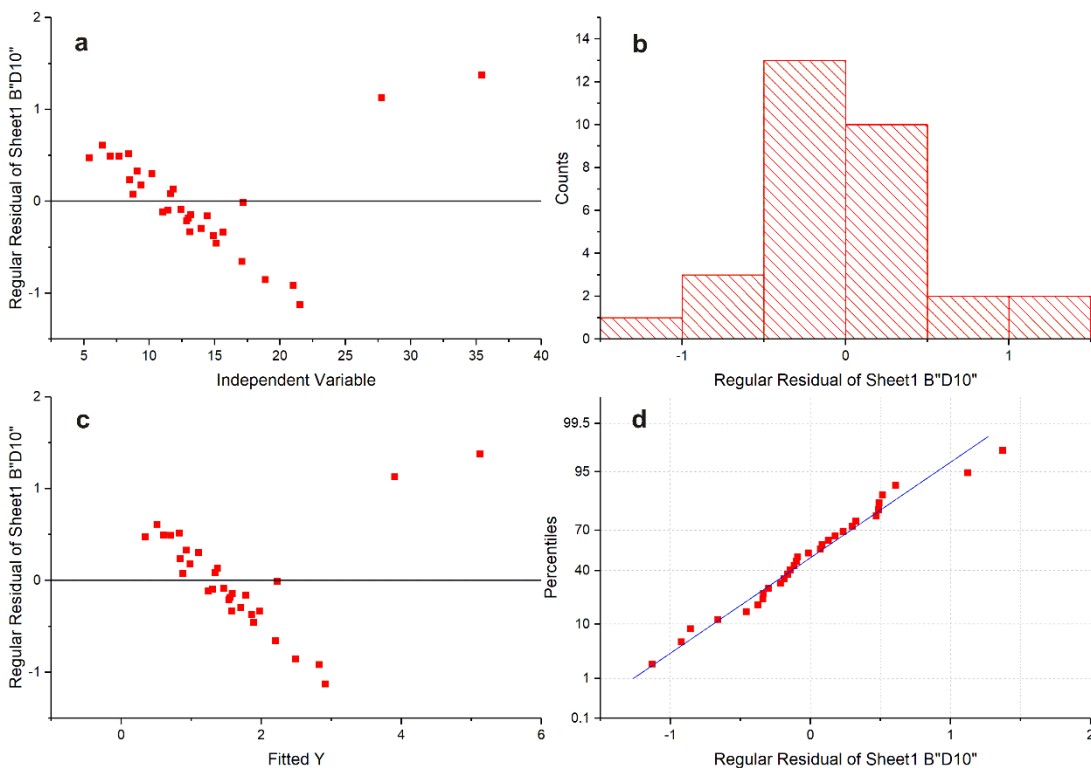

**Figure 5.** Residuals characteristic parameter D10; regional samples: (**a**) shows scatter plot of residuals; (**b**) shows histogram of regular residual; (**c**) shows scatter plot of residuals with fitted Y; and (**d**) shows scatter plot of residual flatten and significance testing.

The dominant clay minerals in the regional soil samples were kaolinite, followed by illite, and some vermiculite, and chlorite (Table 2). Kaolinite content has a weak correlation with rock-forming minerals, such as potassium feldspar and quartz. This finding indicates a nonlinear process of potassium feldspar alteration into kaolinite during granite weathering. Moreover, clay minerals and

quartz were present in the weathering crust over metamorphic sandstone and slate in the Anyuan County area, and their correlation reached 0.68 (Figure 6). This indicates that the conversion ratio of feldspar to clay minerals, after weathering of this metamorphic bedrock, is higher than that of granite in the Longnan County granite.

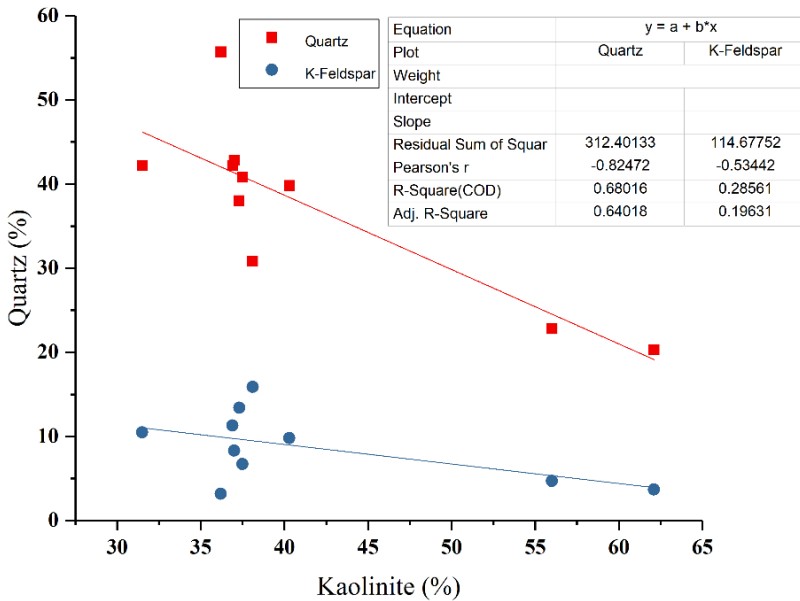

**Figure 6.** Correlation trend of kaolinite, Potassium feldspar, and quartz in samples from Anyuan County.

The relationship between average particle size (Dav) and kaolinite, quartz, and potassium feldspar indicates that the mineral particle size had only a minor effect on the clay mineral content. Quartz and potassium feldspar are mostly distributed on both sides of kaolinite. Quartz is further away from the *X*-axis, while potassium feldspar is closer to the *X*-axis. This distribution, therefore, further confirms that weathering of potassium feldspar has a significant impact on the formation of kaolinite.

Wang et al. (2018) [39] found that the main minerals of the low-grade metamorphic rocks (e.g., meta-sandstone, meta- siltstone, and slate) in the Anyuan County area of southern Jiangxi are 30–70% quartz, 5–30% feldspar, 3–10% biotite, and 3–12% muscovite. The CIA index is in the range of 68–75%. The Yanshanian granite in Longnan County is mainly compounded of 25–32.7% quartz, 31.1–42.4% potassium feldspar, 17–28.9% plagioclase, 3.4–6% biotite, and 1–3.4% muscovite [40]. According to the analysis of major elements in granite in the Zudong mining area, Longnan County [8,41] (Table 6), CIA is in the range of 61–65%, with an average of 63%, which is significantly lower than that of Anyuan County. This suggests that a higher CIA value reflects more extensive loss of $Na^+$, $K^+$, and $Ca^{+2}$ during leaching, enrichment in Al and Si, and more advanced conversion of feldspar to clay minerals [42,43]. This study confirms that the weathering of the parent rock has a significant effect on the formation of soil clay minerals. Weathered feldspar minerals are converted to kaolinites, which are then converted into kaolin minerals under moderate silica and salt-based ion conditions [13]. Furthermore, layered silicate minerals, such as muscovite, biotite, and chlorite, are weathered at varying degrees to form kaolinite minerals [44]. The original rocks of Anyuan County are predominantly metamorphic sandstone, siltstone, and slate, which were found to be relatively broken in the field, and provides favorable conditions for further weathering. As shown by the analysis of kaolinite in Table 1, the average content of kaolinite, illite, and potassium feldspar in Anyuan County (samples AY-01 to AY-12) is 40.66%, 8.03%, and 9.72%, respectively. This indicates that most of the potassium feldspar in the Anyuan County metamorphic bedrock was converted into kaolinite minerals. In Longnan County (samples LN-01 to LN-12), the bedrock is medium-grained granodiorite, with an average kaolinite mineral ratio of 24.67%, i.e., much less than that in Anyuan County.

**Table 6.** Chemical composition (wt %) of Longnan granites (Zudong mining area).

| Sample No. | 1 | 2 | 3 | 4 | 5 | 6 |
|---|---|---|---|---|---|---|
| $SiO_2$ | 70.34 | 75.55 | 76.14 | 74.88 | 72.48 | 74.58 |
| $TiO_2$ | 0.44 | 0.25 | 0.03 | 0.05 | 0.13 | 0.15 |
| $Al_2O_3$ | 14.59 | 12.13 | 12.97 | 13.44 | 13.4 | 13.34 |
| $Fe_2O_3$ | 2.03 | 1.18 | 0.09 | 0.21 | 0.51 | 0.56 |
| FeO | 0.64 | 1.01 | 1.07 | 1.42 | 1.28 | 1.25 |
| MnO | 0.25 | 0.07 | 0.03 | 0.08 | 0.11 | 0.05 |
| MgO | 0.57 | 0.34 | 0.07 | 0.19 | 0.17 | 0.18 |
| CaO | 0.55 | 0.42 | 0.62 | 0.58 | 1.51 | 0.73 |
| $Na_2O$ | 3.37 | 2.74 | 4.25 | 3.97 | 3.62 | 3.65 |
| $K_2O$ | 5.7 | 5.36 | 4.52 | 4.61 | 5.37 | 4.8 |
| $P_2O_5$ | 0.05 | 0.04 | 0.02 | 0.06 | 0.04 | 0.05 |
| LOI | 0.92 | 0.7 | 0.96 | 0.57 | 1.62 | 0.6 |
| Total | 100.33 | 99.85 | 100.77 | 100.28 | 100.24 | 100.03 |
| CIA | 64 | 63 | 63 | 65 | 61 | 64 |

CIA = Chemical Index of Alteration, Data from Bao et al., 2008 [8] and Zhang, 1990 [41].

Differences in kaolinite and illite content between Anyuan and Longnan are not only related to the composition of the parent rock. The sampling depth and topography also have a significant impact on the clay mineral and soil formation [45]. Samples from Anyuan County came from average elevations of between 311 and 468 m, with relatively gentle relief (gradient: 24°–25°) and dense vegetation cover. Owing to the influence of bedrock lithology, geological structure and surface erosion, a concave slope formed in this area, and the weathering profile was deep. Samples from Longnan County, on the other hand, came from average elevations of between 278 and 321 m, from a relief of lightly-weathered residual hills with linear or convex slopes controlled by granite lithology. These nuances of landforms have a crucial influence on parent rock weathering and soil formation.

*5.2. Vertical Variation of Clay Minerals in REE Ores*

An analysis of the Ganxian District soil profile [27] showed that the main clay minerals are kaolinite and illite (Figure 6). Kaolinite content fluctuates from top to bottom, with the lowest content (9.8%) at 115 cm from the bottom of the section. In this section, the layers are mostly located at the bottom of the semiweathered layer, where the granite structure is visible and the weathering degree of the rock is weakened. The content of rock-forming minerals (i.e., quartz, 39.9%; potassium feldspar, 37.9%) in (Table 2) shows that conversion of potassium feldspar to clay minerals was minor, which is consistent with this deeper level of the weathering profile. The peak of kaolinite (35.0%) appears at a depth of 190 cm; the kaolinite content is relatively low in the 130–190 cm interval (14.9-16.5: Table 3). From 20 to 105 cm below surface, the kaolinite content increases irregularly, reaching 28.2% at 105 cm. This increase is probably due to the rapid conversion of feldspar and mica minerals into kaolinite and other clay minerals.

Furthermore, illite content shows variation similar to that of kaolinite, but with sharper changes. Potassium feldspar and quartz also exhibit different variation characteristics. Two distinct horizons are thus resolved in the Ganxian weathering profile can be divided on the basis of clay mineral distribution, as follows (Figure 7).

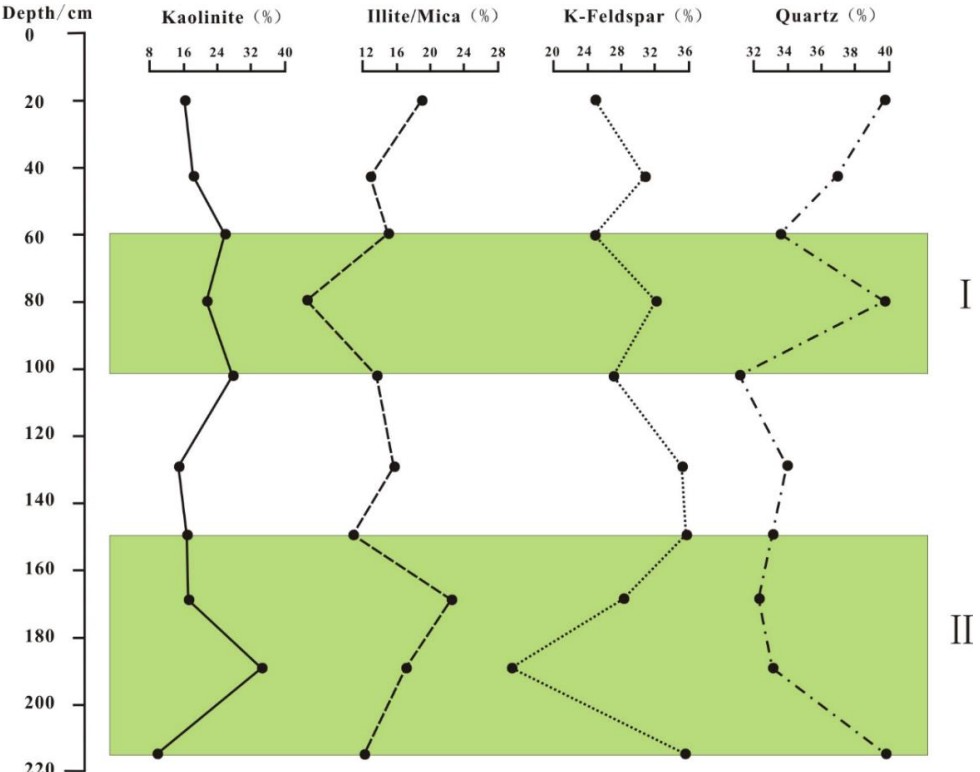

**Figure 7.** Clay and other mineral content in the Ganxian weathering profile.

Band I (60–105 cm): clay mineral content shows a high-low-high variation trend, while the illite content shows a broader range of variation than that of potassium feldspar. The content of quartz, and feldspar shows the opposite trend to that of clay minerals, and the variation range of the quartz content was broader than that of potassium feldspar. Over the course of weathering, clay minerals are converted into kaolinite minerals due to physical and chemical weathering processes. According to Uzarowicz et al. (2011) [46], the composition of clay minerals in the soil surface follows the acidic soil formation process, i.e., it is strictly controlled by the content of chlorite and mica debris, with subsequent conversion of chlorite and mica to montmorillonite and vermiculite. Our analysis shows that kaolinite and illite were reduced concurrently in the profile, while the quartz and potassium feldspar content increased. It is suggested that large quantities of mica and chlorite minerals were formed after weathering and alteration of the parent rock (i.e., granite), thereby controlling the formation and transformation of kaolinite.

An XRD analysis of Sample GX-04-4, showed seven illite-mica diffraction peaks, between d = 4.47 Å and d = 2.50 Å, with a cumulative peak height of 117.8% (Figure 8). The GX-04-1 sample showed a total of four distinct illite-mica diffraction peaks, with a cumulative peak height of 17.4%. XRD analysis further confirmed that, as a result of surface weathering of the granite bedrock, feldspars have altered to chlorite and mica; other clay minerals are due to the weathering and alteration of mica in the original rock. The formation and conversion of kaolinite were limited.

Band II (150–215 cm): clay mineral shows a low-high-low variation trend, opposite to that of band I. Potassium feldspar and quartz show greater changes in an opposite trend. The content of kaolinite and potassium feldspar shows a particularly evident reversal at 190 cm (increase of kaolinite; decrease of potassium feldspar), indicating that the weathering of potassium feldspar contributes to the formation of kaolinite in the soil. In the supergene weathering realm, the conversion of potassium feldspar into kaolinite can be expressed using the following chemical Equation (1) [47]:

$$4KAlSi_3O_8 + 4H^+ + 2H_2O \rightarrow 4K^+ + Al_4Si_4O_{10}(OH)_8 + 8SiO_2 \tag{1}$$

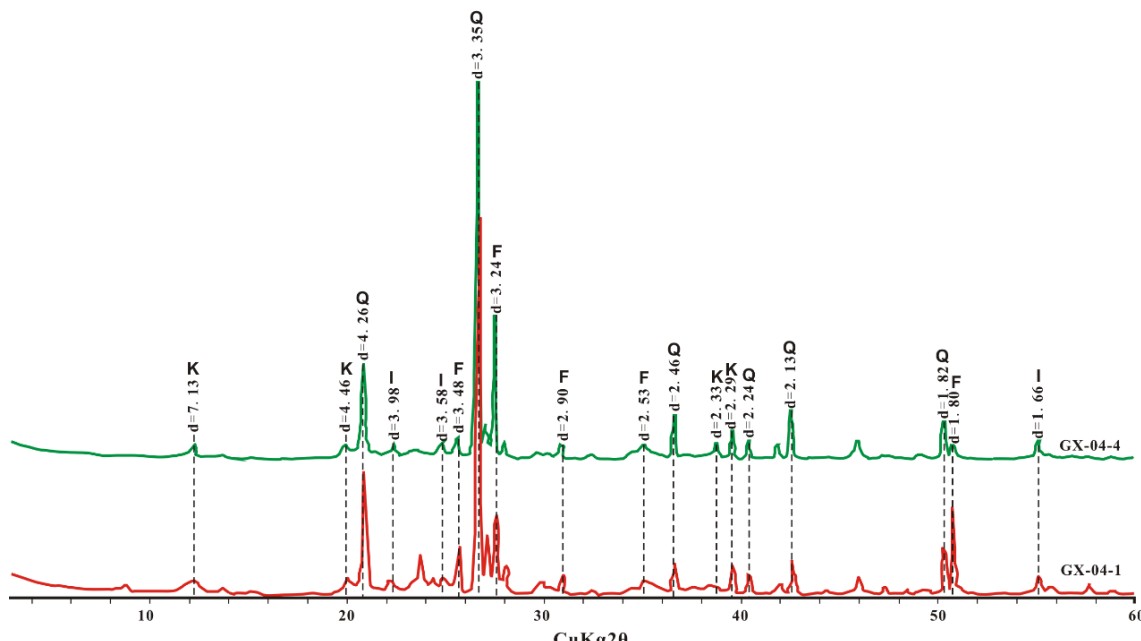

**Figure 8.** XRD spectra of randomly oriented bulk sample from the Ganxian Profile, I: illite-mica, K: kaolinite, Q: quartz, F: feldspars.

Equation (1) shows that decomposition of potassium feldspar releases a large quantity of free metal cations in the soil. This promotes further enrichment in kaolinite. According to Dixon (1989), the formation of kaolinite in the soil requires an adequate amount of silica and a small number of metal cations [48], while the chemical decomposition of potassium feldspar needs to consume $HCO_3^-$ produced from $H^+$ and $CO_2$ [49]. This provides a favorable environment for the formation of kaolinite, forming the distinct volatile characteristic of band II. Moreover, the potassium feldspar diffraction peak in GX-04-1 was significantly reduced as compared to that in GX-04-4 (Figure 9). For quartz, the cumulative diffraction peak height is 195% in Sample GX-04-1, and 79.1% in Sample GX-04-4. This suggests that formation of kaolinite was favored in the $SiO_2$-rich environment.

An analysis of samples collected at different stages of the in situ leaching profile from Longnan County showed that the clay mineral content fluctuated regularly as leaching progressed (Figure 8). In the early stage of leaching, the kaolinite content in the soil was less than 15%, while the illite-mica content was slightly higher than that of kaolinite, fluctuating from 10.1% to 17.2% (Table 3). The potassium feldspar content fluctuated between 16.4%–34.1%, and the quartz content was relatively high. The total kaolinite content increased, and the total potassium feldspar content decreased as leaching progressed (Figure 8). In the later stage of leaching, from the surface to the bottom of soil column, the kaolinite content increased rapidly from 8.9% to 33.0%, then gradually decreased. The illite-mica content also decreased slightly compared to the previous period, with the exception of the bottom of the Longnan soil column, where it increased weakly (from 10.1% to 15%). The content of quartz increased significantly in the upper layer between 46.2%–61.3% and 40.5%–53.65% (Table 3), with an average increase of 32.5%. In the final stage of leaching, the kaolinite content increased significantly: from 22.5% to 31.2% (average growth: 46.2%; Table 3). The illite-mica content increased by between 15% and 20.7%. The potassium feldspar content increased significantly in the final stage of leaching (average growth: 32.9%), while the content of quartz decreased (27.9% to 35.8%).

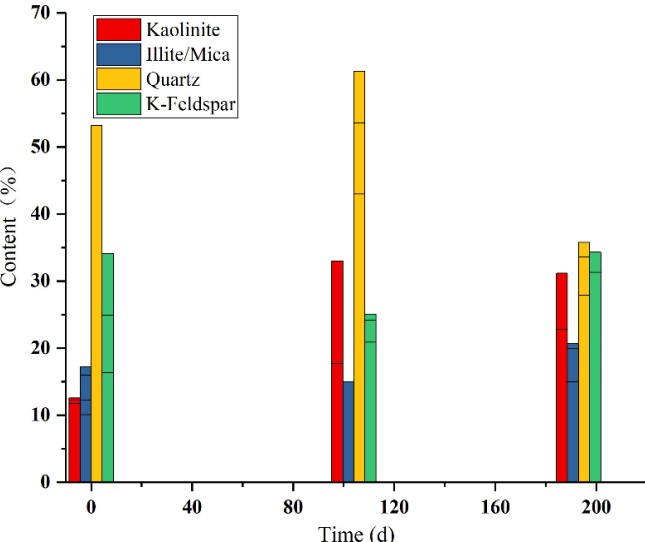

**Figure 9.** Clay content variation during in situ leaching, Longnan Profile.

Parfitt et al. (1983) [20] found that the Si concentration in the soil solution decreased as rainfall increased, reflecting increasing leaching of the soil. Wu et al. (2016) [36] suggested that granite with compact structure has higher strength, a low degree of weathering, and a higher content of residual feldspar. In conditions of sustained weathering by acid rain, cations increased in the leached upper soil, and residual feldspar decomposed rapidly, demonstrating a process of feldspar decomposition, as described by Equation (1). Fine particles of $SiO_2$ gradually migrated from the upper to the lower soil horizons, leading to silica enrichment in the latter.

An analysis of the major clay minerals in different stages of in situ leaching (Figure 10) showed that the vertical migration of clay minerals was significant. Kaolinite did not change significantly in the initial stage of leaching; its content fluctuated between 9.5% and 12.6%, with a standard deviation of 1.22. In the course of leaching, potassium feldspar was consistently weathered and converted to kaolinite [36]. The kaolinite content peaked at a depth of 200 cm, reaching a maximum value of 33%, which is 3.47 times the initial value. During the late leaching stage, the kaolinite content fluctuated slightly. Although it increased slightly from 200 cm, it rapidly reduced afterward. Field investigation in the Longnan profile revealed that about 50 m from the northeast end of the profile, there were rows of 150–180 cm-deep injecting holes along the hill slope (i.e., along 330°–150° direction).

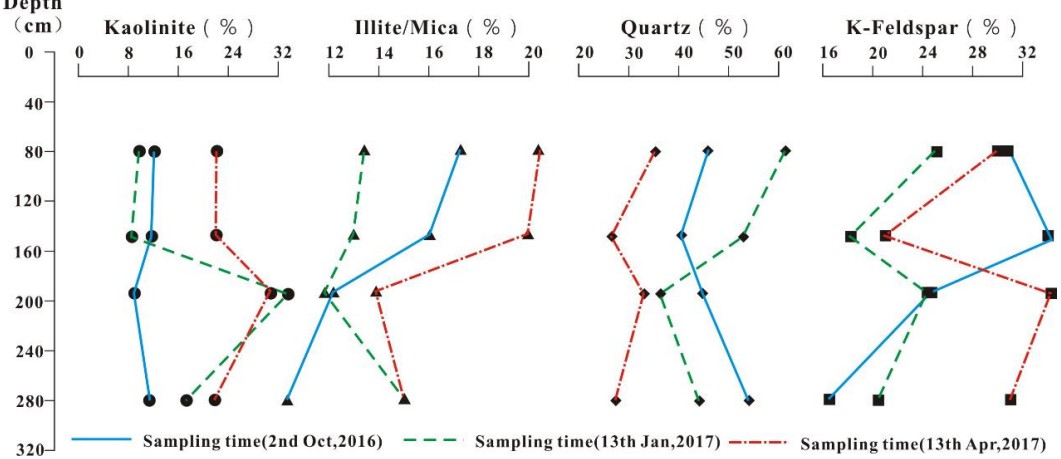

**Figure 10.** Clay mineral, quartz, and potassium feldspar content on the Longnan soil column at different stages of the leaching experiment.

These injection holes received $(NH_4)_2SO_4$ electrolyte solutions over a long period [50]. Due to this, soil in the lower part of the profile was saturated, and its acidity was enhanced, which further enhanced kaolinization of potassium feldspars. In the initial stage of simulated leaching, the illite-mica content was high near the surface of the sediment column (17.2%), and decreased downwards to only 10.1% at the bottom of column. As leaching progressed, the illite-mica content continued to decrease with increasing depth in the 0–200 cm interval; from 200 cm downwards, however, it increased significantly in the later stage of leaching. The range of illite-mica contents was similar to that of the leaching process, but higher than that in the leaching process. Different clay minerals have different geochemical behavior, and may have different physical and chemical responses to factors such as pH, salinity, and blocking cations [51]. Previous studies have shown that illite forms from potassium feldspar alteration in two different types of microsystems [14]: a) in the early stages of the weathering process, along crystal joints of orthoclase with muscovite or biotite; and b) in the final stages of weathering, where the original structure of the parent rock is destroyed. In both cases, illite forms in association with other clay minerals, i.e., smectite in the early weathering stage, and kaolinite in the late weathering stage. With the exception of its interlayer charge and consequent potassium content, illite is, in many ways, similar to phengite mica [14]. As indicated above, this depth presented an increase in the illite-mica contents due to potassium feldspar weathering and dissolution, silicon release, and hydrated layers mineral formation.

Comparing kaolinite with illite-mica, no correlation between the two was evident in the initial stage of in situ leaching, while a significant negative correlation was present in the later stage of leaching. This trend indicates that, besides the decomposition of potassium feldspar into kaolinite, a large quantity of interlayer silicates such as illite-mica are converted to kaolinite as leaching progresses.

The change in quartz and potassium feldspar content became complex with increasing depth in the soil column (Figure 10). In the early leaching stage, the quartz content initially decreased, then increased, while potassium feldspar content showed the opposite trend. For both minerals, the inflection points were at a depth of 150 cm. Between 80 and 150 cm in the soil column, the content of both quartz and potassium feldspar decreased as leaching progressed. However, quartz inherited the characteristics of the initially decreasing trend, i.e., downwards from 195 cm, it turned into an increasing trend. In contrast, potassium feldspar showed the opposite trend downwards from 150 cm. In the later leaching stage, both quartz and potassium feldspar showed a decreasing-increasing-decreasing trend with depth, but the variation range of potassium feldspar was broader than that of quartz.

Although the content of vermiculite was not measured in the simulated leaching experiment, regional sampling of REE ore revealed low vermiculite content (Table 2). Vermiculite formation occurs in two stages: a) in the early stage, the common mica weathering products are dioctahedral vermiculites whose layer charge is lower than that of the parent mica; b) in the second stage, mica dissolution advances further, and corroded zones of polyphase assemblage of dioctahedral hydroxy-vermiculite appear within mica crystals [14]. Vermiculite has good ion-adsorption properties; its adsorption capacity of REE ions is nearly 0.2 mmol/g. Vermiculites adsorbing REE ions can be regenerated by cation ion-exchange reagents according to the following reaction [52]:

$$
\begin{aligned}
&\left\{(Mg, Fe, Al)_6[(Si, Al)_8O_{20}](OH)_4\right\}_m \cdot nRE^{3+} \cdot eH_2O + 3nM^+ \\
&\quad = \left\{(Mg, Fe, Al_6)[(Si, Al)_8O_{20}](OH)_4\right\}_m \cdot 3nM^+ \cdot eH_2O + nRE^{3+}
\end{aligned}
\tag{2}
$$

Chemical Equation (2) shows that the decomposition of potassium feldspar releases large quantities of free Al, Fe and Mg cations, and Si in the soil. This favors the formation of vermiculite (Figure 10). However, in the in situ leaching profile, the vermiculite content was low, probably due to the flow of leaching liquid flow and surface water elution.

Previous studies indicated that ion-absorbed REE ores mainly contain halloysite, illite, and kaolinite, and less smectite [1–3,9]. The factors that strongly favor the formation of smectite include low-lying topography, poor drainage, and base-rich parent material, leading to chemical conditions of

high pH, high silica activity, and an abundance of basic cations [13]. In leaching conditions with lower pH, as in our leaching experiment, it was impossible to form abundant smectite. With the exception of vermiculite transformed to smectite, the original REE ores in our area of study contained less than 1% smectite [3].

### 5.3. Simulating Migration of Clay Minerals during Leaching

A total of eight simulated soil columns were subjected to different experimental conditions of pH, immersion concentration, and leaching rate. A high-acidity leaching mining solution was used to further decompose the remaining feldspar in the soil to clay minerals [17]. The simulated leaching experiment showed that kaolinite was further enriched in the soil column. The leaching solution concentration and leaching rate also had an effect the rate of decomposition of silicate minerals in the soil columns [53].

In the T1 soil column (Figure 11), the kaolinite content was initially high; subsequently, it decreased and then increased slowly as leaching progressed. The most prominent diffraction peak of kaolinite (d = 7.20 Å in Figure 10) is relatively weak in the middle part of the soil column, compared with the upper and lower parts. Similar results have been reported from the weathering profiles of other Mesozoic granites [54]. Other relatively prominent kaolinite diffraction peaks were at d = 2.33 and 1.99 Å. As shown in Table 3, the variation was also weaker in the middle of the soil column compared with the upper and lower parts. Kaolinite and illite-mica had similar diffraction peak characteristics across the soil column. For illite-mica, the most prominent initial diffraction peak corresponded to d = 10.01 Å, and the (002) crystal planes showed significantly high diffraction. Other evident peaks were at d = 5.0 Å and d = 4.46 Å. As the diffraction angle increased, strong diffraction peaks appeared at d = 2.44 Å, and d = 1.99 Å. Quartz showed a prominent diffraction peak for d = 4.26 Å (peak height: 2615; diffraction intensity: 21.9%). High quartz diffraction peaks at d = 3.34 Å were present in all samples. Potassium feldspar presented the first evident diffraction peak for d between 6.6 and 6.45 Å (corresponding to a diffraction angle (2-Theta) at between 13.4° and 13.7°. This finding reflects the different diffraction intensities of different crystal faces. Significant diffraction peaks were also present at d = 3.24, 2.28, and 1.98 Å.

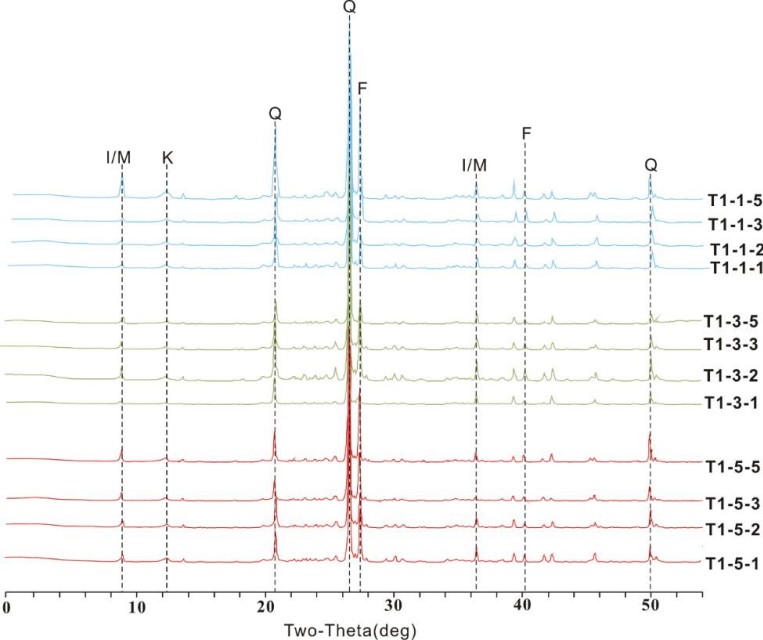

**Figure 11.** XRD spectra of randomly oriented bulk samples from Soil Column T1 (simulated leaching). I/M: illite-mica; K: kaolinite; Q: quartz; F: potassium feldspars.

Here, we discuss clay mineral content under three key conditions of simulated leaching.

① Same concentration of leaching solution and leaching rate; different pH values (soil columns T1–T3 in Table 3, Figure 12):

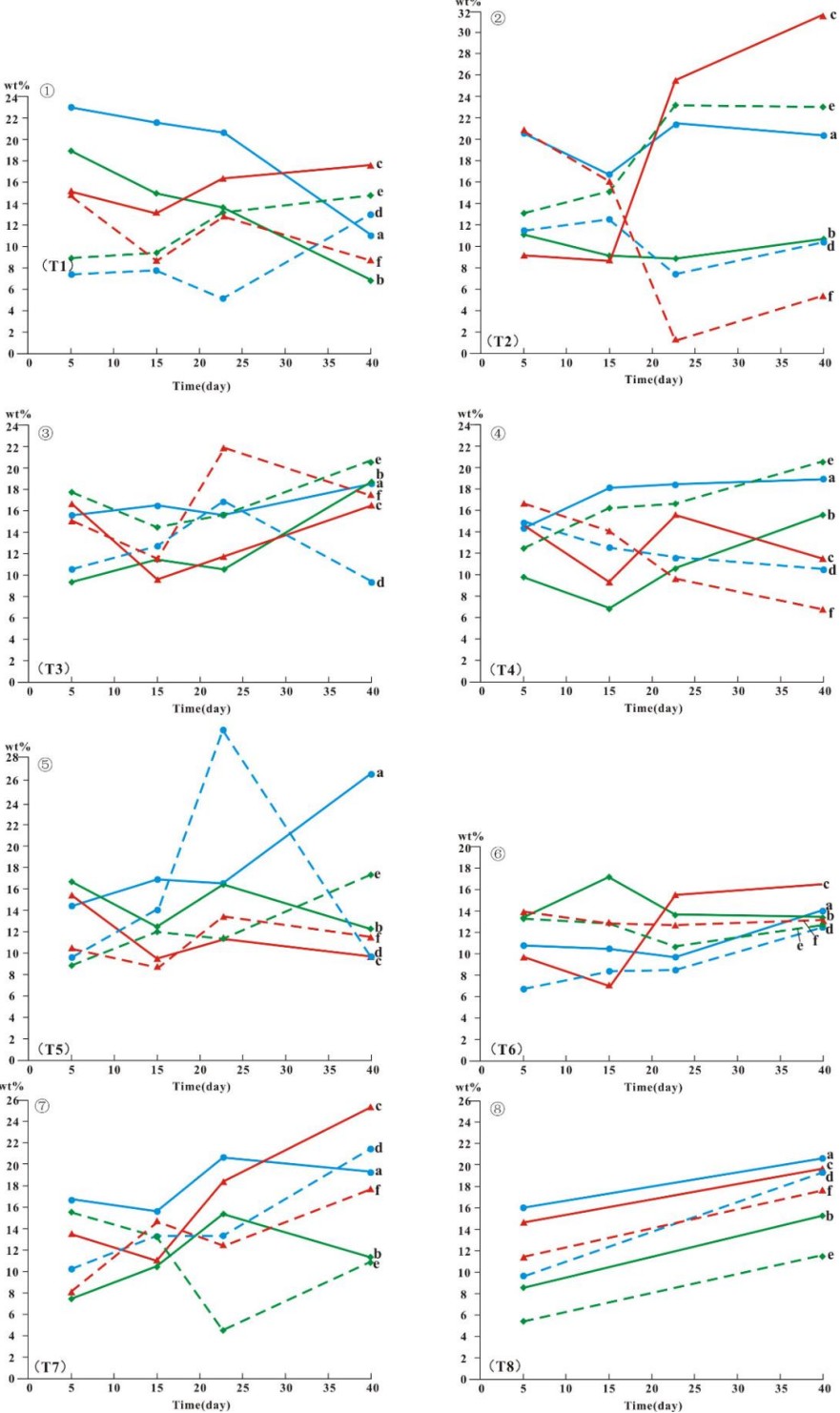

**Figure 12.** Changing clay mineral content in different conditions of simulated leaching. (**a–c**) kaolinite content sample at sample locations of 30, 70, and 110 cm, respectively; (**d–f**) illite-mica content at sample locations of 30, 70, and 110 cm, respectively.

With the concentration of leaching solution and rate of leaching being stable, the migration and enrichment of clay minerals is controlled by pH. Soil column T1 shows that as leaching progresses, the kaolinite content decreases at depths of 30 and 70 cm in the soil column. After two weeks of leaching, the soil column was gradually enriched in the clay minerals, and the content of kaolinite content in the bottom layer increased (① in Figure 12); the lower the pH, the more favorable the soil conditions for the decomposition of parent rock, particularly for the hydrolysis of feldspars and the formation of clay minerals. As indicated above, the soil column T2 was leached with solution of pH = 3, the lowest pH value in this group. After 15 days of leaching, the kaolinite content in soil Column T2 increased markedly, reaching a maximum of 31.5% (② in Figure 12). In the final leaching stage, the kaolinite content increased significantly at a depth of 110 cm, indicating that REEs were adsorbed and released due to the high recovery rate. Other studies have demonstrated that the higher the pH of the leaching solution, the higher the adsorption capacity of clay minerals for rare earth ions. In a weak acid environment (pH = 4), the kaolinite content in different layers of soil Column T3 soil increased slowly (③ in Figure 12).

② Same pH and leaching rate; different concentration of the leaching solution (soil Columns T4, T5):

In soil Columns T4 and T5, the kaolinite content increased gradually in the near-surface layer (30 cm) as leaching progressed (③ in Figure 12 and ④ in Figure 12. The higher the concentration of the leaching solution, the flatter the change in clay mineral composition. After 15 days of leaching, the concentration of the leaching solution in soil Column T5 decreased to 4% $(NH_4)_2SO_4$, while the clay mineral content near the surface increased rapidly, eventually reaching 26.7%. At a depth of 70 cm, Soil Columns T4 and T5 showed different responses: with the lower-concentration leaching solution of T5, the kaolinite content in this layer decreased slowly, with an average decrease of about 25%; with the more concentrated leaching solution, i.e., T4, the kaolinite content showed increasing volatility. At a depth of 110 cm, different leaching concentrations produced similar effects on the kaolinite content.

At the same time, illite-mica minerals exhibited different responses under different leaching conditions. The high-concentration leaching solution in soil Column T4 gradually reduced the illite-mica content. On the 23rd day, with the 4% $(NH_4)_2SO_4$ solution, the illite-mica content increased abruptly, further verifying the enhanced adsorption capacity of illite in a weakly acidic environment. As indicated above, at the upper level of the soil columns, clay minerals were concentrated by the high-concentration leaching solution, but at the bottom of the soil columns, this was not observed (⑤ in Figure 12). Our experiment suggested that the effective concentration of leaching solution was 8%.

③ Same pH and leaching concentration; different leaching rates:

Soil Columns T6 and T7 showed that the leaching rate had an impact on the content of kaolinite (⑥ in Figure 12, ⑦ in Figure 12). At 30 cm, the content of kaolinite in soil Column T6 increased slowly as leaching progressed. After 15 days, the kaolinite content increased significantly in soil column T7; in soil column T6, at 70 cm, it increased slightly, and gradually decreased as leaching progressed. In soil column T7, after 20 days of leaching, the total kaolinite content increased, as it did in the upper layer (30 cm), demonstrating that clay minerals have similar structures in REE ore. At a depth of 110 cm, the kaolinite content decreased after 15 days in columns T6 and T7, and then increased again. This change probably reflects the translocation of fine-grained kaolin minerals from the upper and middle to the lower parts of the profile as leaching progresses. The longer the leaching time, the higher the content of fine-grained clay minerals at the bottom of profile; some of these even clog the porosity, which reduces the flushing out of rare-earth ions and hampers leaching [36]. In the leaching conditions of this sample group, with a leaching rate of 5 mL/min, enrichment in clay minerals was at a rate of over 1 mL/min. This indicated that a high recovery rate is not possible at a slower rate of leaching.

A comparison between different simulated leaching conditions showed that the content of kaolinite and other clay minerals tended to increase from the initial stage until the completion of leaching. The fluctuation of clay mineral content is the result of the combination of different pHs, leaching concentrations, and leaching rates (⑧ in Figure 12). However, it cannot be assumed that a certain

leaching condition determines the outcome of the leaching mining process. Leaching mining is a complex chemical process, and the variation of clay mineral content only reflects one aspect of it. It is impossible to adequately simulate ore leaching conditions in sediment columns, in view of the boundary restrictions of a soil column, the horizontal flow of ore leaching solution, and the ore texture and structure. For these reasons, our experimental results were not as expected, although the leaching conditions in our soil column experiment were controlled. However, comparing the results experimental leaching of soil Columns T2, T4, T6, and T7 at 110 cm after 23 days of leaching (Figure 11) with the results of in situ leaching in the Longnan section (Figure 9), in both cases, the kaolinite content increased while the illite-mica content decreased. We expect that simulated leaching experiments applying many different leaching conditions will permit us to explore how various factors influence clay mineral fluctuation during leaching.

The REE distribution on the kaolinite-water interface is considered to be the result of the adsorption of REE ions by kaolinite, and is strongly controlled by pH [55]. Tian et al. (date) [52] found that the REE recovery was up to over 96% with a $NH_4^+$ concentration in the raffinate solution of 0.2 g/L and a pH of 2. Other research suggested if the pH of the leaching agent is either too high or too low, recovery of REE is reduced. The optimal pH values were between 4 and 8. The maximum leaching efficiency of REE was 91% [56]. In our experimental study, the pH values were between 3 and 5. Disregarding other external factors, we suggest that REE recovery of over 90% can be achieved through leach mining with a leaching solution of 8 wt% concentration and a pH of 5, at a leaching rate of 5 mL/min.

## 6. Conclusions

Analyses of REE ore clay mineral properties and grain size, and the distribution and variation of clay minerals, quartz, and feldspars in in situ leaching and simulated leach mining led to the following conclusions:

(1) In the surveyed REE mine areas, the soil particle size (i.e., volume frequency) curve showed unimodal and bimodal distribution. Many cumulative particle size distribution curves had a "B" shape, with particle sizes of 3.74–30.46 μm. Other curves showed an "A" shape, which indicates that the increase of coarse particles in the soil affects particle gradation. An analysis of particle size eigenvalues showed that D10 was 0.82–5.03 μm, while D90 ranged from 11.99 to 60.99 μm, with a standard deviation of 11.44. These values reflect discreteness among the coarse particle fraction. Taking the Dav parameter as the independent variable and other characteristic parameters as the dependent variables of the regression analysis, the regression coefficient order was D90 > D [3,4] > D50 > D [2,3] > D10. This finding revealed that the increase of surface average particle size in soils has a greater effect on coarse particles. A D10 residual analysis showed that with an increase of the average particle size of Dav content, the volume fraction below 10% was reduced. The residual error showed a linear relationship with a 99.5% confidence interval.

(2) A regional clay mineral analysis showed that ion-absorbed REE ores formed on different bedrock lithologies have similar clay mineral contents. The main clay minerals were kaolinite, illite, chlorite, and vermiculite. The kaolinite content ranged from a minimum of 8.8% to a maximum of 62.1%. In the granite weathering area of Longnan County, the kaolinite content had weak correlation with the content of potassium feldspar, quartz, and other rock-forming minerals. The chemical weathering index (CIA) was in the range of 61–65, and the average kaolinite content was low. In the metamorphic terrain of Anyuan County, the kaolinite content was strongly correlated with quartz, and the CIA ranged from 68 to 75. This reveals a higher degree of feldspar weathering and conversion to clay minerals in metamorphic bedrock.

(3) Studying the natural weathering profile of granite in the Ganxian District showed that the content of kaolinite was relatively low at 60–105 cm below surface. There, conversion of feldspars to chlorite and mica in the course of weathering, and the increased content of mica in the parent rock, limited the formation of kaolinite. At 150–215 cm below the surface, the kaolinite content increased.

This was attributed to the desiliconization of overlying feldspar minerals, which increased the $SiO_2$ content in the weathering mantle and promoted the decomposition of feldspars into kaolinite.

(4) An analysis of in situ leaching mining profile showed that in the early stage of leaching, the content of kaolinite in the soil is relatively low, i.e., lower than that of illite-mica. As the leaching progressed and potassium feldspar continued to convert to kaolinite, significant kaolinite peaks appeared in the profile, and the illite-mica content became slightly reduced. This indicates that interlayer silicate minerals such as illite-mica were progressively converted to kaolinite. In a later stage of leaching, the kaolinite content became slightly reduced in comparison with that of the earlier leaching stage, and the content of illite-mica and potassium feldspar increased.

(5) Simulated leaching mining reveals that under the same conditions of leaching solution concentration and leaching rate, the migration and enrichment of clay minerals were controlled by pH; the lower the pH value, the more favorable the conditions for clay mineral formation. With the same pH and leaching rate, but different concentrations of the leaching solution, a high-concentration leaching solution (i.e., 12% $(NH_4)_2SO_4$)) resulted in a slow increase in the kaolinite content in the upper part of the soil profile (30 cm). Under stable pH and immersion concentration conditions, the variation of the leaching rate influences soil formation and clay mineralization processes in different horizons within the soil profile. Based on these results, and disregarding other external factors, we suggest that REE recovery of over 90% can be achieved through leach mining with a leaching solution of 8% concentration and a pH of 5, at a leaching rate of 5 mL/min.

**Author Contributions:** Conceptualization and methodology, L.C., and H.C.; investigation, L.C., X.J., L.Q., and H.C.; experimental analysis, X.J.; writing-review and editing, L.C. and Z.H.; partial writing, X.J., and L.Q.; plotting, H.C.; software, H.D. All authors have read and agreed to the published version of the manuscript.

**Funding:** This project was funded by the National Science Foundation of China (Grant no. 41967038), China Postdoctoral Science Foundation (Grant no. 2015M582530), and the Science Foundation of Jiangxi Provincial Department of Education (Grant no. GJJ150657).

**Acknowledgments:** We are indebted to Kaixing Wu and Tao Sun for their helpful discussions and suggestions. We are also sincerely thanks for Houston–Oil and gas laboratory and the Key Laboratory of Nonferrous Metal Materials Science and Engineering of the Ministry of Education. The authors are grateful to two anonymous reviewers for their helpful suggestions and comments. We want to thank Editage for the English language editing.

**Conflicts of Interest:** The authors declare no conflict of interest.

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
