# Peer review of "Grain Size Distribution and Clay Mineral Distinction of Rare Earth Ore through Different Methods"

_minerals, doi:10.3390/min10040353_

Round 1

Reviewer 1 Report

The revised manuscript on the study on the grain sizes distribution and clay minerals distinction of rare earth ore through different methods has been improved from the first review. However, the language must be significantly improved, as many discussions and sentences are very hard to follow, and the intended meaning is unclear. The article requires a thorough English language review in order to make it readable, and to ensure that the desired results are being conveyed properly to the reader. At the moment, it is difficult in some parts to understand the intended meaning of the authours. Some particularly problematic sections are highlighted in the attached file, and a number of other queries and questions are provided for the authours to address.

Author Response

Dear reviewer,

    We thank you so much for improving our manuscript to propose comment and suggestion. Now, we have revised all the unclear sentences and answered the question from the reviewer of highlighting. The manuscript language has been improved  by an English language review.

    Best regards

    Lingkang  Chen

Reviewer 2 Report

I mostly accept the answers provided in the document Response to reviewer -2. However, I think that the deleted Figure 1 did have value for the reader and should be restored. Also there is no justification given why the original figures 2 and 3 were removed. These figures also would be insightful for the readers. The previously mentioned points reflect the box of "Are the results clearly presented" where I ticked the option "must be improved".

The newly added sections need English editing.

Typing error: line 45 -> REEs, not RREs

Author Response

Dear reviewer,

We thank you so much for improving our manuscript to propose comment and suggestion. Now, we have revised and added the figure 1(The location of sampling sites), in this figure, we have highlighted the sites of

in-situ leach mining profile and simulated leaching profile, but have not still supplemented figure 2 and figure 3 of old version’s. Our authors consider that the sampling sites of topographical features and sampling rules had been revealed by figure 1 and section of ‘3.1. Sample collection’ statement. Moreover, the profile simulated leaching experiment and experimental facility had been expressed with as much particular as possible by ‘3.2. Simulated leaching experiment’ statement. Due to highlight the clay minerals of rare earth ore and its grain size distribution, simplify the structure of the article, and according to others editorial comment, we have only added figure 1. Please criticise us for anything inappropriate.

We have revised all the unclear sentences and answered the question from your highlighting. The manuscript language has been improved by an English language review.

Best regards,

Lingkang CHEN.

Round 2

Reviewer 1 Report

The manuscript was thoroughly revised considering the suggestions from the previous review. I have found the changes made are acceptable and the manuscript is in a good form for publication. I have made minor edits for final suggestions.

Author Response

Dear reviewer,

We have revised our manuscript and proposed all the comment and suggestion from you. We are using the "Track Changes" function in Microsoft Word, and thank you for your help so many times.

Best regards

Lingkang Chen

This manuscript is a resubmission of an earlier submission. The following is a list of the peer review reports and author responses from that submission.

Round 1

Reviewer 1 Report

The paper by Chen et al. presents the results of clay mineral analyses for ion adsorbed rare earth ores, and characterizes their grain size distribution, clay mineralogy, and leaching behaviours for regional soil surveys, in situ leaching at weathered crust based mine, and lab-scale simulated leaching. While much of the work done is of interest and represents a coordinated effort, significant improvement is required to clarify the main results and discussions of the research. For example, many tables of data are presented but are not particularly useful, while graphs illustrating up to 27 datasets in different colors are shown. Meanwhile, the results from the lab-scale simulated leaching experiment are somewhat dismissed as being non-representative of field conditions, and so their merit for inclusion in this paper must be carefully reconsidered. In order to meet the standards required for publication, I suggest the authors carefully curate the work that is presented and focus the scope of the study to draw out the major conclusiosn based on sound scientific support and evidence from the data collected. As it stands, the purpose, objective, presentation and interpretation of the data, as well as clear achievements of the study are not well defined or constrained. In addition, the authours should consult a list of suggested changes in the attached PDF, as a starting point to try to focus the paper.  

Reviewer 2 Report

First of all the authors are to be congratulated for writing a very professional article. The paper "Grain Sizes Distribution and Clay Minerals Distinction of Rare Earth Ore through Different Methods" presents an original study that concerns the matters of clay minerals in ion-absorbed rare earth element deposits. Such deposits and their exploitation is currently highly relevant globally. At the same time, not all the aspects of the leaching and behaviour of the ion-absorbed REE deposits under different conditions are fully understood.
Current study examines the REE deposits by combining different methods and approaches: grain size distribution, distinction of clay minerals (and their parent minerals), leaching behaviour in a natural deposit as well as a in an experimental soil profile. The sampling and experimental design seems to be well representing the actual conditions, and the information about the leaching collected over several years time is especially valuable.
The very minor comments are written on the manuscript file. The only comments for improving the already very good manuscript would be the following:
(1) the authors might add some parameters (grade and tonnage of REEs) of the rare earth deposits in the introductory section just for a general background;
(2) the usage of the term "rare earth minerals" could be substituted with some alternative phrasing such as "rare earth associated minerals", because the current wording could
be potentially misleading (line 472, line 547)
(3) line 251: it is not clear where this information originates from
(4) some figures should be improved for better clarity: Figure 1, improve the text quality; Figure 7: add indication for sub-figures a-d: Figure 15: add indication for sub-figures;
A follow-up study could discuss the results of this study with REE geochemistry and leaching behaviour. The authors mentioned that they suspected the presence of kaolinite and halloysite
that are hard to distinguish. As they mentioned themselves, it could be beneficial to analyse some of the samples in scanning electron microscopy.

As a conclusion the present paper will be a valuable contribution for ion-absorption REE deposits literature as well as in general in the context of soil science and aspects of weathering processes.
